# Molecular responses in abdominal subcutaneous adipose tissue after a session of endurance exercise: effects of exercise intensity

Cheehoon Ahn[1,2] [ID], Tao Zhang[1] [ID], Thomas Rode[1], Gayoung Yang[1] [ID], Olivia K. Chugh[1], Sierra Ellis[1], Sophia Ghayur[1], Shriya Mehta[1], Ryan Salzman[1], Hui Jiang[3] [ID], Stephen C. J. Parker[3,4,5], Charles F. Burant[6] [ID] and Jeffrey F. Horowitz[1] [ID]

[1] *Substrate Metabolism Laboratory, School of Kinesiology, University of Michigan, Ann Arbor, Michigan, USA*

[2] *Translational Research Institute, AdventHealth, Orlando, Florida, USA*

[3] *Department of Biostatistics, University of Michigan, Ann Arbor, Michigan, USA*

[4] *Gilbert S. Omenn Department of Computational Medicine & Bioinformatics, University of Michigan, Ann Arbor, Michigan, USA*

[5] *Department of Human Genetics, University of Michigan, Ann Arbor, Michigan, USA*

[6] *Division of Metabolism, Endocrinology, and Diabetes, Department of Internal Medicine, University of Michigan, Ann Arbor, Michigan, USA*

Handling Editors: Karyn Hamilton & Jørn Helge

The peer review history is available in the Supporting Information section of this article (https://doi.org/10.1113/JP289339#support-information-section).

**Abstract figure legend** An acute session of aerobic exercise induces robust transcriptional changes in abdominal subcutaneous adipose tissue 1.5 h after exercise. Exercise at low-, moderate- and high-intensities each reveals some distinct molecular responses in adipose tissue.

**Abstract** The primary aim of this study was to compare the acute effects of three exercise intensities on abdominal subcutaneous adipose tissue (aSAT) transcriptome in regular exercisers. A total of

This article was first published as a preprint. Ahn C, Zhang T, Rode T, Yang G, Chugh OK, Ellis S, Ghayur S, Mehta S, Salzman R, Jiang H, Parker SCJ, Burant CF, Horowitz JF. 2025. Acute session of three endurance exercise intensities alters subcutaneous adipose tissue transcriptome in regular exercisers. https://doi.org/10.1101/2025.05.02.651890

45 adults who exercise regularly were assigned to perform a single session of either low-intensity continuous (LOW; 60 min at 30% VO$_2$max; $n = 15$), moderate-intensity continuous (MOD; 45 min at 65% VO$_2$max; $n = 15$) or high-intensity interval exercise (HIGH; 10 × 1 min at 90% VO$_2$max interspersed with 1 min active recovery; $n = 15$). aSAT biopsy samples were collected before and 1.5 h after the exercise session for bulk RNA sequencing and targeted protein immunoassays. HIGH upregulated genes were involved in angiogenesis, protein secretion and insulin signalling pathways, whereas MOD and LOW upregulated genes regulated extracellular matrix (ECM) remodelling, ribosome biogenesis and oxidative phosphorylation pathways. Exercise-induced changes in aSAT clock genes, ERK protein phosphorylation and circulating cytokines were similar after all three exercise treatments. Network analysis identified exercise-responsive gene clusters linked to cardio-metabolic health traits. Cell-type analysis highlighted a heterogeneous response of aSAT cell types to exercise, with distinct patterns observed across exercise intensities. Collectively, our data characterize early responses in aSAT after a single session of exercise. Because adaptations to exercise training stem from an accrual of responses after each session of exercise, these early responses to exercise are likely important contributors to the long-term structural and functional changes that occur in adipose tissue in response to exercise training.

(Received 16 June 2025; accepted after revision 22 September 2025; first published online 13 October 2025)

**Corresponding author** J. F. Horowitz: Substrate Metabolism Laboratory, School of Kinesiology, University of Michigan, Ann Arbor, MI, USA.    Email: jeffhoro@umich.edu

### Key points

- Chronic adaptations in adipose tissue from regular exercise support cardiometabolic health, but the acute molecular triggers of these adaptations remain unclear.
- We show that acute exercise alters gene expression, along with ERK phosphorylation in adipose tissue of regular exercisers.
- Exercise intensity shapes the transcriptomic response: high-intensity exercise induces inflammatory, cytokine and genes, whereas lower intensities upregulate genes involved in protein translation and oxidative phosphorylation.
- Network and cell-type analyses highlight intensity-specific adipose responses, revealing gene modules linked to health traits and differential engagement of adipocyte subpopulations.

## Introduction

The cardiometabolic health benefits of regular exercise are well documented (Knowler et al., 2002; Myers, 2003); however the precise mechanisms driving these effects remain incompletely understood. Most studies examining the metabolic benefits of exercise largely focus on adaptations in skeletal muscle, but far less is known about exercise-induced adaptations in adipose tissue that also underlie some of the health benefits of exercise (Ahn et al., 2022, 2024). Exercise training has been found to increase abdominal subcutaneous adipose tissue (aSAT) capillarization and mitochondrial proteins, remodel extracellular matrix (ECM) and lower pro-inflammatory macrophage infiltration (Ahn et al., 2022, 2024). However how exercise triggers these adaptive

**Cheehoon Ahn** received his BS in physical education from Seoul National University in Korea and completed his MS and PhD in movement science at the University of Michigan. He is currently a postdoctoral fellow at the Translational Research Institute at AdventHealth, where he focuses on translational metabolism. His research integrates high-dimensional approaches – including single-cell RNA sequencing, spatial transcriptomics and multi-omic bioinformatics – to investigate human adipose tissue biology in the contexts of obesity, exercise and ageing.

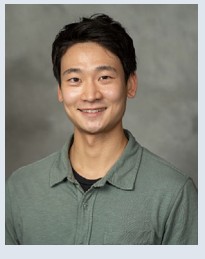

responses in aSAT, and whether the exercise intensity and/or energy expended during exercise may impact these triggers, is unclear.

Adaptations to regular exercise typically result from the accrual of repeated exposure to acute transcriptional changes that occur shortly after each exercise session (Perry et al., 2010). For example the increase in skeletal muscle mitochondrial density in response to weeks or months of endurance training is largely due to the repetitive transient increase in transcriptional regulators of mitochondrial biogenesis and transcripts involved in oxidative phosphorylation that occur in the hours after each exercise training session (Egan & Zierath, 2013; Mahoney et al., 2005; Pilegaard et al., 2003). The increase in aSAT mitochondrial density, which has also been found with exercise training (Brandao et al., 2019; Riis et al., 2019), may occur through upregulation of the similar transcriptional pathways as observed in skeletal muscle (Matta et al., 2021; Sutherland et al., 2009). Additionally we previously observed a robust increase in the mRNA expression of the key angiogenic transcription factor vascular endothelial growth factor A (VEGFA) 1 h after aerobic exercise (Van Pelt et al., 2017), which might be part of the initial step for enhanced vascularization in aSAT observed with exercise training (Ahn et al., 2022). Therefore, identifying transcriptomic signals after a session of exercise is critical for understanding the mechanisms underlying adaptations in aSAT that occur with exercise training. Importantly, many of the well-described adaptations to exercise training vary depending on the intensity, duration and/or energy expended during the exercise training sessions. This is also true for at least some of the adaptations in adipose tissue. For example, it has been reported that high-intensity training induced some more pronounced adaptations in adipose tissue (e.g. greater abundance of capillaries and smaller adipocytes and less pro-inflammatory macrophage infiltration) compared to moderate-intensity training (Khalafi et al., 2020; Kolahdouzi et al., 2019).

The primary aim of this study was to compare the acute effects of exercises performed at low-, moderate- and high-intensity that are commonly prescribed for health and fitness on aSAT transcriptome. We hypothesized that (1) a single session of exercise would alter aSAT transcriptome, cytokine production and protein phosphorylation, and (2) the exercise-induced effects would be distinct among three different exercise sessions.

## Methods

### Ethical approval

Written informed consent was obtained from all participants before the study. This study conformed to the standards set by the Declaration of Helsinki, except for registration in a database, and was approved by the University of Michigan Institutional Review Board (reference no.: HUM00204857). This study is registered at clinicaltrials.gov (NCT05365334).

### Participants

We recruited only participants who regularly engage in endurance exercise to avoid the confounding influence of the stress response of an exercise stimulus in persons who are not accustomed to exercise. This approach is supported by prior findings showing that the number of differentially expressed genes in aSAT dropped by over 90% (from 3882 to 349 genes) when previously sedentary adults repeated the same acute exercise protocol after 6 weeks of training (Fabre et al., 2018). Out of 190 participants who expressed interest in the study 45 healthy participants who met our eligibility criteria for body mass index (BMI 20–30 kg/m$^2$), age (18–40 years), regular recreational exercise ($\geq$30 min, $\geq$3 days/week, moderate- to vigorous-intensity endurance-type exercise for at least 2 months), and reported having stable body weight for at least 6 months, were enrolled and completed the study (Fig. 1*A*). Of the 45 participants 42 engaged in either running, cycling or both (28 runners, 3 cyclists and 11 who did both). The remaining three participants participated in tennis, kickboxing or other forms of cardiovascular exercise. Thirteen participants also incorporated recreational-level resistance training into their routines. As part of the eligibility criteria participants were also not taking any medications or supplements known to affect their metabolism, except for contraceptive medications for some female participants. All participants were free of cardiovascular and metabolic disease. Female participants were confirmed to be premenopausal and not pregnant or lactating at the time of enrolment. All subjects completed a detailed medical history survey and resting electrocardiogram, which were reviewed by a physician before any testing.

### Group assignment

Participants were assigned to one of three exercise treatment groups using a manual counterbalancing approach. As each participant enrolled we reviewed their age, sex, BMI, body weight and VO$_2$peak, and then assigned them to a group to ensure that distributions of key demographic and physiological variables (e.g. sex, age, BMI, body weight, VO$_2$peak) remained as balanced as possible across groups. This process was done iteratively at the time of enrolment to avoid large imbalances in baseline characteristics. The three exercise treatment groups were as follows: (1) low-intensity exercise (LOW, 60 min of steady-state exercise at ~30% VO$_2$peak,

$n = 15$); (2) moderate-intensity exercise (MOD, 45 min of steady-state exercise at ~65% VO$_2$peak, $n = 15$); high-intensity exercise (HIGH, 10 × 1 min intervals at ~90% VO$_2$peak with 1 min of active recovery between intervals, $n = 15$).

## Preliminary procedures

**Body composition.** Body composition was determined using dual-energy X-ray absorptiometry (Lunar DPX DEXA scanner, GE, WI).

**Aerobic capacity (VO$_2$peak).** Graded exercise testing was performed on a stationary cycle ergometer (Corvival, Lode, Netherlands) using an incremental exercise protocol starting at 40 W and increasing 20 W per minute until volitional exhaustion. The rate of oxygen consumption was measured throughout this test using a metabolic cart (Quark CPET, COSMED, Italy), and VO$_2$peak was determined as the highest 30 s average before volitional fatigue. Measurements of respiratory exchange ratio (RER) $\geq$ 1.2 and maximal heart rate (HR$_{max}$) $\geq$ 90% of age-predicted values (i.e. 220-age) were used as secondary indices to help confirm maximal effort during these tests. The experimental exercise sessions were performed on the same cycle ergometer as VO$_2$peak was measured.

**Exercise familiarization exercise.** All participants underwent at least one supervised familiarization exercise session during which they performed the exercise protocol consistent with their group assignment. This familiarization exercise session was completed at least 4 days before the experimental trial.

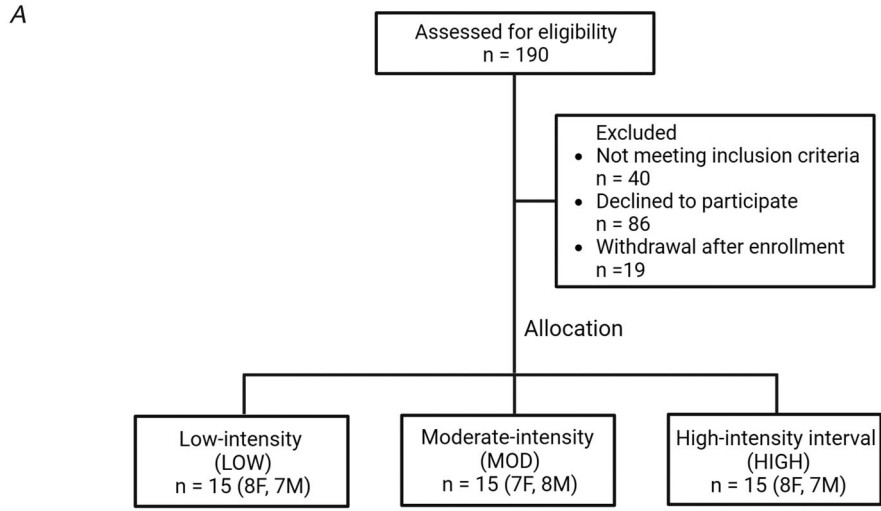

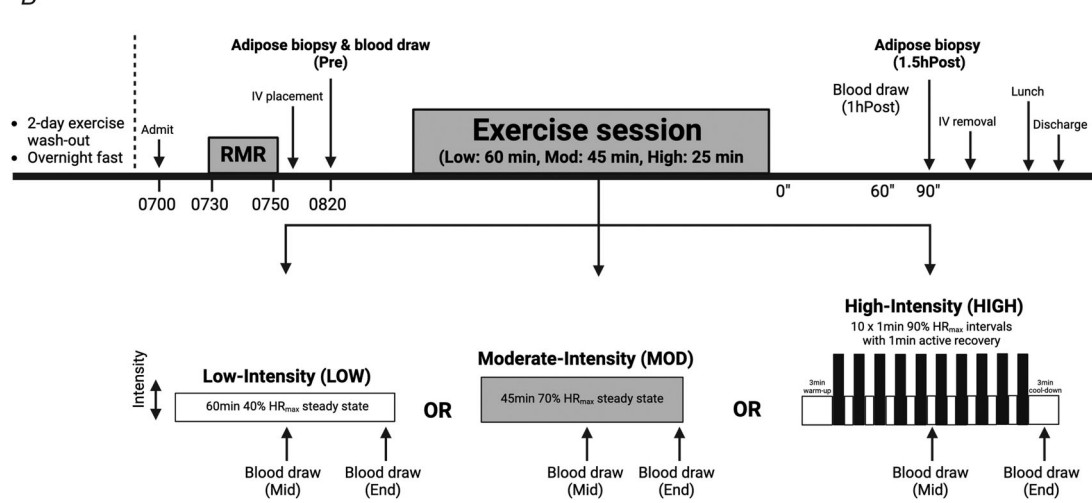

**Figure 1. CONSORT diagram and study design**
*A*, CONSORT diagram of the study. *B*, schematic of study design.

## Experimental trial

The experimental trial is outlined in Fig. 1*B*. All participants were instructed to refrain from structured physical activity for 48 h prior to the experimental trial. The evening before the trial, participants were instructed to eat a standardized dinner (30% of estimated total daily energy expenditure) at ∼19:00 h and snack (10% of estimated total daily energy expenditure) at ∼22:00 h. The next morning participants arrived at the clinical facility at 07:00 h after an overnight fast. After 30 min of resting quietly, resting metabolic rate was measured using indirect calorimetry (TrueOne 2400, ParvoMedics, Salt Lake City, UT, USA). An intravenous catheter was then inserted into the antecubital vein on one arm for blood collection. At ∼09:00 h an aSAT sample was collected by aspiration 5–10 cm distal to the umbilicus (either left or right). aSAT samples were snap-frozen in liquid nitrogen and stored at −80°C for later quantification of transcriptome (RNA sequencing) and protein abundance (targeted immunoblots) (see details below). A pre-exercise blood sample was collected in conjunction with the aSAT biopsy. Subjects then performed their prescribed exercise session.

Participants assigned to LOW performed steady-state exercise at 40% heart rate reserve (HRR) (∼30% $VO_2$peak) (Swain & Leutholtz, 1997) for 60 min (Fig. 1*B*); subjects assigned to MOD performed steady-state exercise at 70% HRR (∼65% $VO_2$peak) for 45 min (Fig. 1*B*); and subjects assigned to HIGH performed 10 × 1 min intervals at 90% HRR (∼90% $VO_2$peak) with 1 min of active recovery between intervals (Fig. 1*B*). The HIGH exercise session also included a 3 min warm-up and cool-down before and after the interval protocol (total exercise time in HIGH was 25 min). Exercise intensity was monitored by a telemetry heart rate device (Polar, Finland).

Blood samples were collected at the mid-point (MID) of each exercise session (at ∼30' in LOW, at ∼22' in MOD and between fifth and sixth intervals for HIGH). Additional blood samples were collected during the final 30–60 s of exercise (End) and 1 h postexercise (1 hPost). Blood samples were centrifuged at 2000 *g* at 4°C for 15 min. Serum and plasma were aliquoted and stored at −80°C until analysis for circulating factors. After exercise all subjects rested quietly on a bed for 1 h and 30 min. The postexercise aSAT biopsy samples (Post) were collected in the same manner as the pre-exercise samples but on the opposite side of the umbilicus.

## Analytical procedures

**RNA sequencing.** RNA was extracted from 100 mg of frozen aSAT by using a Qiagen Mini RNA extraction kit (74104, Qiagen, Redwood City, CA). Bulk RNA-seq was conducted at the Oklahoma Medical Research Foundation (OMRF) Clinical Genomics Centre. Prior to RNA-seq

analysis quality control measures were implemented. Concentration of RNA was ascertained via fluorometric analysis using a Thermo Fisher Qubit 4 fluorometer. Overall quality of RNA was verified using an Agilent Tapestation 4200 instrument. After initial QC steps mRNA was isolated, and libraries were generated using the IDT xGen RNA kit (IDT Cat# 10009814) according to the manufacturers protocol. Briefly mature mRNA was enriched from total RNA via pull-down using beads coated with oligo-dT homopolymers, using the NEB NEBNext Poly(A) mRNA Magnetic Isolation Kit (NEB Cat#E7490L). The mRNA molecules were then chemically fragmented, and the first strand of cDNA was generated using random primers incorporating a truncated i5 adapter sequence. After a bead-based cleanup the 3' end of the single-stranded cDNA was ligated to a truncated i7 adapter. Libraries were then indexed using IDT xGen Unique Dual Indexing primers (IDT Cat# 10012162). Final libraries for each sample were assayed using the Agilent Tapestation 4200 for appropriate size and quantity. These libraries were then pooled in equimolar amounts as ascertained using fluorometric analyses. Final pools were absolutely quantified using quantitative PCR (qPCR) on an ABI QuantStudio6 instrument with NEB Illumina Library Quantification reagents. Sequencing of the libraries was performed using an Illumina NovaSeq X Plus with PE150 reads. Average reads per sample were 28.5 million reads. The paired-end RNA-seq reads were mapped to human transcripts annotated in GENCODE v44 (Harrow et al., 2012) using Bowtie (Langmead et al., 2009). Gene-level read counts were estimated using rSeq (Jiang & Wong, 2009; Salzman et al., 2011).

## Bioinformatics

*Differential analysis and gene set testing.* Normalization of gene count data and differential analysis were performed using DESeq2 (Love et al., 2014). Raw gene-level counts were normalized using the median-of-ratios method. The design accounted for repeated measures (paired pre- and post-exercise samples) and group-specific effects using the formula ∼ group + group:ID + group:condition, where group represents exercise treatment group, ID is the subject identifier and condition indicates time point. Genes with low expression were filtered out prior to analysis by retaining only those with ≥10 counts in at least 15 samples irrespective of group assignment. Wald tests were used to test for differential expression between pre- and post-exercise conditions within each group (e.g. contrast = list ('groupHIGH.conditionpost')) and between groups (e.g. contrast = list('groupHIGH.conditionpost', 'groupLOW.conditionpost')). *P*-values were adjusted using the Benjamini-Hochberg method. Genes with an adjusted $P < 0.05$ were considered significant. For gene

set testing reference gene sets were combined from Gene Ontology, Kyoto Encyclopedia of Genes and Genomes (KEGG) and HALLMARK, obtained using the hypeR R package. Competitive gene set test accounting for inter-gene correlation (CAMERA) (Wu & Smyth, 2012) from limma R package was used on all genes, using signed log (*P*-value). BH-adjusted *P*-value < 0.05 was used as a cut-off for the determination of significantly enriched gene sets. Only statistically significant (adjusted *P* < 0.05) terms are shown in Fig. 4*F–H* and *L–N*.

*Weighted gene co-expression network analysis.* Weighted gene co-expression network analysis (WGCNA) (version 1.73) (Langfelder & Horvath, 2008) was employed to identify distinct, non-overlapping clusters (modules) of aSAT transcripts at baseline (*n* = 45 samples). Connectivity for each transcript was calculated by summing its correlation strengths with all other transcripts. A scale-free signed network was constructed using an optimal soft-threshold power ($\beta = 6$). Using a dynamic tree-cutting algorithm and a merging threshold of 0.3, 16 distinct modules were identified, including Module 0. Module 0, often referred to as the 'grey' module, comprises transcripts that could not be assigned to any other module due to weak correlation patterns. Top hub genes within each module were determined based on their kME values (module eigengene-based connectivity). Overrepresentation analysis (ORA) was conducted on module member genes using the hypeR R package. Module eigengenes (MEs), representing the first principal eigenvector of each module, were extracted, and their relationships with clinical traits and secreted protein abundances were evaluated using biweight midcorrelation (Langfelder & Horvath, 2012). The coefficient, bicor, is similar to Pearson's correlation coefficient r, but it produces more robust outcomes by accounting for outliers. Postexercise ME was acquired by recalculating the first principal component using the same member genes identified from the pre-exercise modules. Biweight midcorrelation was used to correlate the change of ME ($\Delta$, from pre-exercise to 1.5 h postexercise) and change of circulating factor concentrations ($\Delta$, from pre-exercise to end of exercise). Paired *t* test was used to assess ME differences between pre-exercise and 1.5 h postexercise within each group.

*Cell-type enrichment and deconvolution.* Cell-type marker gene lists were curated from three previously published single-nuclei (sn) RNA-seq (Emont et al., 2022; Whytock et al., 2024) or spatial transcriptomics (Backdahl et al., 2021) datasets of human adipose tissue. Although all datasets assess mRNA abundance, we acknowledge the differences in transcript coverage. snRNA-seq captures the nuclear transcriptome, which can differ from cytoplasmic or whole-cell mRNA profiles, whereas spatial transcriptomics typically measure both nuclear and cytoplasmic mRNA. Despite these methodological differences recent work has shown that snRNA-seq data can effectively capture key cell-type–specific transcriptional programmes (Emont et al., 2022; Massier et al., 2023). For gene markers obtained from Emont et al. and Whytock et al. we applied a logFC > 1 threshold to retain genes that were significantly and specifically expressed by each cell type. No expression-level filter was applied to markers from Backdahl et al., as gene expression from spatial transcriptomics is inherently low. The curated markers for each cell type were treated as independent gene sets. Statistical gene set testing was performed using CAMERA. Bubble plots were generated using TMSig R package. Deconvolution analysis was conducted using a specialized pipeline designed to estimate cell-type proportions (%) in aSAT (Ahn, Divoux, et al., 2025). This pipeline utilizes a gene signature matrix derived from the snRNA-seq dataset reported by Whytock et al. (Whytock et al., 2024), which achieved the highest gene coverage per nucleus for aSAT to date, enabled by full-length transcriptomics. Only snRNA-seq data from younger adults (18–40 years, *n* = 10) were used to closely match the age range of subjects in this study.

**Targeted immunoblots.** We used capillary electrophoresis-based western blot (JESS, ProteinSimple, San Jose, CA, USA) to measure the abundance of proteins of interest in aSAT lysates. A portion of each aspirated adipose biopsy sample (∼90 mg wet weight) was homogenized in ice-cold 1X RIPA buffer (89901, Thermofisher) with freshly added protease and phosphatase inhibitors (P8340, P5726, and P0044; Sigma) using two 5 mm steel beads (TissueLyser II, Qiagen, Redwood City, CA, USA). Homogenates were rotated at 50 rpm for 60 min at 4°C and then centrifuged at 4°C for 3 × 15 min at 15,000 *g*. Supernatants were transferred to new tubes after each centrifugation to reduce lipid contamination. Protein concentration of the cleared supernatant was assessed using the bicinchoninic acid method (#23225, Thermofisher). Samples were prepared in 4× Laemmli buffer and heated at 95°C for 7 min. Equal amounts of protein (0.16 µg) were mixed with Simple Western sample buffer and fluorescent mix and loaded into capillaries of the 12–230 kDa JESS separation module, 25-capillary cartridges (SW-W003). All experiments were performed on the automated JESS device in accordance with the manufacturer's instructions. Protein abundance was normalized to total protein staining. Statistical analysis was performed in SPSS (IBM) using a two-way mixed-effects ANOVA with time and group as fixed effects followed by least significant difference *post hoc* testing. Primary antibodies used were hormone-sensitive lipase (HSL, #18381, Cell Signalling Technology), phospho-HSL (Ser565) (pHSL[S565], #4137,

Cell Signalling Technology), phospho-HSL (Ser660) (pHSL$^{S660}$, #4126, Cell Signalling Technology), protein kinase B (AKT, #9272, Cell Signalling Technology), phospho-AKT (Ser473) (pAKT$^{S473}$, #9271, Cell Signalling Technology), phospho-AKT (Thr308) (pAKT$^{T308}$, #13038, Cell Signalling Technology), p38 mitogen-activated protein kinase (P38, #9212, Cell Signalling Technology), phospho-P38 MAPK (Thr180/Tyr182) (pP38$^{T180/Y192}$, #9211, Cell Signalling Technology), p44/42 MAPK extracellular signal-regulated kinase (ERK, #4695, Cell Signalling Technology), phospho-p44/42 MAPK (Thr202/Tyr204) (pERK$^{T202/Y204}$, #4376, Cell Signalling Technology), signal transducer and activator of transcription 3 (STAT3, #12640, Cell Signalling Technology) and phospho-STAT3 (Tyr705) (pSTAT3$^{Y705}$, #9145, Cell Signalling Technology).

**Blood measurements.** Plasma concentrations of glucose (TR-15221, ThermoFisher), plasma fatty acids (NC9517309, NC9517311; Fujifilm Medical Systems) and plasma glycerol (F6428, Sigma) were assessed using commercially available kits. Plasma lactate was measured as previously described (Hohorst, 1965). Serum insulin concentration was assessed using a chemiluminescent immunoassay (Siemens 1000). Epinephrine and norepinephrine (Abnova, Taipei City, Taiwan; KA1877) and cortisol (R&D Cat. no. KGE008B) were measured using enzyme-linked immunosorbent assay (ELISA). Plasma concentrations of interleukin 1$\beta$ (IL1$\beta$), IL10, IL6, interferon $\gamma$ (IFN$\gamma$) and tumour necrosis factor $\alpha$ (TNF$\alpha$) were measured using customized Luminex Multiplex kit (HSTCMAG-28SK). For some targets (IL6 and IL10) the measured concentration was below the detection threshold in some participants (IL6: 5 LOW, 1 MOD, 2 HIGH; IL10: 1 LOW, 1 HIGH) at least in one time point. Missing values from these participants were imputed with the lowest detected level for each cytokine.

**Plasma volume correction.** A single session of aerobic exercise induces changes in the concentration of many circulating factors (i.e. metabolites or hormones) (Chow et al., 2022). However the true effect of exercise on inducing the bona fide synthesis of these factors can be confounded when blood volume is not accounted for, because haemoconcentration during exercise leads to reduction in plasma volume (Dill & Costill, 1974). Therefore we measured plasma calcium (Ca) concentration as the marker for haemoconcentration (Alis et al., 2015). The concentration of circulating parameters at Mid, End, 1 hPost and 1.5 hPost was corrected for plasma volume as follows: $[parameter]_c = [parameter]_u/(1 + \Delta Ca(\%)/100)$, where $\Delta Ca(\%)$ refers to the percentage change of Ca concentration relative to pre-exercise (Pre)

level, and c and u sub-indices denote corrected and uncorrected concentration, respectively.

**Histology.** Only pre-exercise aSAT samples were analysed for histology. Core biopsies were fixed in 10% formalin for 48 h immediately following collection and then paraffin-embedded using a Sakura Tissue Tek TEC system. Tissues were sectioned at 5 μm thickness using a Leica RM2235 microtome and mounted onto microscope slides (ThermoFisher #109508-WH). Slides were deparaffinized and rehydrated through xylene and graded ethanol (100%, 95%, 70%) washes. To reduce variability staining was performed in batches, and microscopy settings, such as exposure time, were kept consistent across all samples.

*Sirius Red and fat cell size.* For assessment of collagen deposition and adipocyte size, sections were stained with Picro Sirius Red (#36 554-8, Sigma-Aldrich) for 1 h following deparaffinization. Stained slides were imaged at 10× magnification using a Keyence BZ-X700 widefield microscope. Adipocyte area was quantified using the AdipoQ plugin for ImageJ (Sieckmann et al., 2022).

*Adipose tissue macrophage immunohistochemistry.* To evaluate macrophage content sections underwent antigen retrieval in 0.5 mм HCl-glycine buffer (pH 3.0) at 90°C for 20 min, followed by blocking with 3% hydrogen peroxide in methanol (15 min), streptavidin (15 min), biotin (15 min) and 5% normal goat serum (1 h). Primary antibodies (MRC1/CD206, #MAB25341, R&D Systems; CD14, #114R-14, Sigma-Aldrich) were applied overnight at 4°C. The following day, slides were incubated with biotinylated secondary antibody (ThermoFisher #NC9801827, 75 min) followed by HRP-linked streptavidin (ThermoFisher #S911, 45 min). Tyramide signal amplification (TSA555 for CD206, #B409533; TSA647 for CD14, #B40958, both Thermo-Fisher) was used for 10 min to enhance signal. GS-lectin was applied for membrane visualization, and nuclei were counterstained with Hoechst 33342 (Invitrogen #H3570). Slides were mounted using ProLong Gold and imaged at 10× magnification with a Keyence BZ-X700 fluorescence microscope. Macrophage abundance (CD206+ and CD14+ cells per adipocyte) was quantified using ImageJ.

### Statistical analyses

Baseline participant characteristics were stratified by intervention group and sex, resulting in six comparison groups. A two-way ANOVA was applied to each continuous variable to assess the main effect of group, sex and their interaction. The between-group comparisons were conducted using the aov() function, with the grouping factor specified as Group_Sex (e.g.

HIGH_F, MOD_M, etc.). For *post hoc* analyses pairwise comparisons were performed using the emmeans R package, applying Tukey's honestly significant difference correction for multiple comparisons. To evaluate the effects of time point and group on each outcome variable (excluding RNA-seq) two-way linear mixed-effects models were fit using the lmer() function from the lme4 R package. Each model included a fixed effect for the time point * group interaction and a random intercept for participant (ID) to account for repeated measures. The reference level for time point was set to Pre using relevel() to aid interpretability of contrasts. Type III ANOVA tests for fixed effects were conducted using the Anova() function from the car R package, appropriate for unbalanced designs with interactions. *Post hoc* comparisons were performed using the emmeans package, where estimated marginal means and pairwise contrasts across time point * group combinations were extracted (emmeans(model, pairwise ~ time point * group)). Statistical significance was defined as $P < 0.05$. Analyses were performed using R (R, Vienna, Austria), and data are presented as mean $\pm$ SD.

## Results

### Subject characteristics and heart rate responses during exercise

A total of 45 participants (15 LOW, 15 MOD and 15 HIGH) completed the study. All participants were healthy, without obesity, and all were regular exercisers (Table 1). As designed there was no difference in baseline anthropometric characteristics or aerobic fitness (VO$_2$peak) among groups (Table 1). Participants exhibited a normal range of metabolic health indices as evidenced by relatively low fasting plasma insulin ($5.8 \pm 2.9$ mU/l), glucose ($4.9 \pm 0.5$ mmol/l), fatty acid ($307 \pm 151$ μmol/l) and triglyceride concentrations ($0.8 \pm 0.4$ mmol/l), all of which did not differ among groups (Table 1). As expected body weight and lean mass were higher in males compared to females in each group ($P < 0.001$) (Table 1). During the steady-state exercise sessions average HR was $105 \pm 7$ bpm during LOW and $140 \pm 9$ bpm during MOD (representing $40 \pm 5\%$ and $69 \pm 6\%$ HRR, respectively). During HIGH average HR during the 1 min high-intensity intervals was $161 \pm 13$ bpm ($86 \pm 8\%$ HRR and $90 \pm 5\%$ HR$_{max}$).

### Concentrations of circulating factors

Plasma concentrations of all circulating factors measured in our study were corrected for the rapid reduction in plasma volume that occurs at the onset of exercise, using calcium as a marker of this exercise-induced haemoconcentration (Alis et al., 2015) (Fig. 2*A*). Plasma calcium concentration was increased in the middle of exercise (i.e. MID time point) in all groups, which remained elevated at End time point for HIGH and MOD ($P < 0.001$ and $P = 0.039$, respectively) but not for LOW ($P = 0.236$) (Fig. 2*A*). Plasma calcium concentration was significantly higher in HIGH compared to LOW at End time point ($P = 0.011$). As anticipated HIGH and MOD significantly increased plasma epinephrine and norepinephrine concentrations above Pre (for epinephrine $P < 0.001$ and $P = 0.0016$ for HIGH and MOD, respectively; and for norepinephrine $P < 0.001$ for both groups), and concentrations of both hormones were more than twofold greater in HIGH compared to LOW and MOD at the end of exercise (all $P < 0.001$), with no difference between LOW and MOD ($P = 0.999$) (Fig. 2*B* and *C*). There was a trend towards increased epinephrine levels during LOW ($P = 0.055$), whereas norepinephrine levels remained unchanged ($P = 0.218$). Similarly both HIGH and MOD significantly increased plasma lactate concentrations above Pre (both $P < 0.001$), with the highest concentrations found after HIGH ($P < 0.001$) *vs.* MOD and LOW; Fig. 2*D*). Plasma lactate concentration during MOD was also significantly greater than LOW ($P < 0.001$) (Fig. 2*D*). Plasma concentrations of catecholamine and lactate returned to pre-exercise levels 1 hPost in all groups. In contrast we did not detect a significant change in plasma cortisol concentration during exercise in any of the groups (main effect of time, $P = 0.215$; group, $P = 0.953$; time*group, $P = 0.573$) (Fig. 2*E*).

As expected plasma glycerol concentration increased during exercise compared to Pre in all groups (main effect of time $P < 0.001$ for both Mid and End), and concentrations declined to near pre-exercise levels 1 hPost (Fig. 2*F*). There were no significant differences in changes of plasma glycerol concentrations among groups (time*group, $P = 0.222$) (Fig. 2*F*). Plasma fatty acid concentration was significantly elevated after exercise in LOW ($P < 0.001$ and $P = 0.0021$ for End and 1 hPost, respectively) (Fig. 2*G*). During MOD plasma fatty acid concentration was not significantly elevated above pre-exercise and remained less than LOW after exercise ($P = 0.0014$ and $P = 0.0362$ for End and 1 hPost, respectively) (Fig. 2*G*). In contrast to LOW and MOD, HIGH reduced plasma fatty acid concentration during exercise ($P = 0.0398$), and fatty acid concentration 1 hPost in HIGH was significantly lower than both MOD ($P = 0.0136$) and LOW ($P < 0.001$) (Fig. 2*G*). Plasma insulin concentration was slightly, yet significantly, elevated at 1 hPost in HIGH ($P = 0.0250$) (Fig. 2*H*), whereas glucose concentrations were unaltered by all exercise groups (main effect of time, $P = 0.586$; group, $P = 0.762$) (Fig. 2*I*). Exercise induced a modest increase in plasma triglyceride concentration (main effect of time; $P = 0.0210$ and $P = 0.0160$ for Mid and End, respectively)

**Table 1. Baseline subject characteristics and circulating metabolic biomarkers**

| | Low | | Mod | | High | | Group diff |
|---|---|---|---|---|---|---|---|
| | F (*n* = 8) | M (*n* = 7) | F (*n* = 7) | M (*n* = 8) | F (*n* = 8) | M (*n* = 7) | NS |
| Age (years) | 29.8 (6.6) | 33.1 (6.0) | 29.9 (7.3) | 31.4 (5.8) | 32.8 (5.5) | 27.6 (7.7) | NS |
| Height (cm) | 165.9 (2.8) | 178.5 (6.5)** | 168.4 (5.4) | 180.1 (7.5)** | 165.7 (5.0) | 180.2 (6.5)*** | NS |
| Bodyweight (kg) | 67.1 (5.3) | 79.2 (8.0)* | 66.3 (7.7) | 81.8 (7.1)** | 66.6 (8.5) | 82.4 (7.4)** | NS |
| Lean mass (kg) | 45.0 (3.9) | 59.0 (4.3)*** | 43.8 (3.5) | 61.0 (7.4)**** | 44.4 (4.2) | 62.1 (7.4)**** | NS |
| Fat mass (kg) | 19.3 (3.2) | 15.0 (8.0) | 19.7 (5.8) | 15.1 (8.3) | 19.5 (6.3) | 17.5 (8.6) | NS |
| BMI (kg/m$^2$) | 24.4 (2.2) | 24.8 (2.4) | 23.4 (2.4) | 25.2 (2.0) | 24.2 (2.6) | 25.4 (2.6) | NS |
| Body fat (%) | 29.9 (3.8) | 21.6 (5.2) | 30.6 (6.4) | 22.3 (7.0) | 30.0 (6.3) | 21.6 (8.7) | NS |
| SBP (mmHg) | 116 (8) | 119 (8) | 121 (9) | 126 (8) | 117 (12) | 122 (9) | NS |
| DBP (mmHg) | 69 (8) | 75 (3) | 78 (6) | 77 (8) | 69 (8) | 74 (9) | NS |
| VO$_2$peak (ml/min) | 2327 (227) | 2905 (245) | 2460 (339) | 3062 (694) | 2475 (423) | 3354 (283)** | NS |
| VO$_2$peak (ml/kg/min) | 34.9 (4.2) | 37.0 (5.0) | 37.4 (5.7) | 37.4 (7.3) | 37.2 (4.9) | 41.1 (5.8) | NS |
| VO$_2$peak (ml/kg LM/min) | 52.0 (5.6) | 49.3 (4.1) | 56.2 (5.8) | 49.8 (5.9) | 55.5 (6.1) | 54.6 (7.3) | NS |
| RMR (kcal/day) | 1316 (110) | 1566 (86) | 1300 (168) | 1580 (239) | 1420 (240) | 1647 (199) | NS |
| FatOX (µmol/min) | 330 (73) | 377 (83) | 291 (66) | 393 (87) | 339 (75) | 392 (62) | NS |
| Fasting glucose (mmol/l) | 4.8 (0.7) | 5.0. (0.6) | 4.9 (0.4) | 5.2 (0.5) | 4.7 (0.6) | 5.0 (0.4) | NS |
| Fasting insulin (mU/l) | 6.8 (1.9) | 5.8 (3.4) | 5.2 (1.5) | 5.6 (3.3) | 6.1 (4.3) | 5.1 (2.4) | NS |
| Fasting fatty acids (µM) | 449 (219) | 261 (126) | 356 (81) | 221 (128) | 324 (104) | 224 (87) | NS |
| Fasting triglycerides (mM) | 0.80 (0.27) | 0.73 (0.31) | 1.01 (0.39) | 0.77 (0.42) | 0.79 (0.26) | 0.90 (0.53) | NS |
| Fat cell size (µm$^2$) | 1479 (272) | 1780 (437) | 1539 (458) | 1642 (478) | 1445 (170) | 1662 (352) | NS |
| CD14 MΦ per adipocyte (count) | 0.08 (0.03) | 0.06 (0.04) | 0.06 (0.03) | 0.07 (0.06) | 0.06 (0.03) | 0.05 (0.03) | NS |
| CD206 MΦ per adipocyte (count) | 0.06 (0.02) | 0.05 (0.02) | 0.05 (0.05) | 0.04 (0.02) | 0.05 (0.01) | 0.06 (0.02) | NS |
| Sirius Red (%) | 0.11 (0.02) | 0.11 (0.01) | 0.12 (0.07) | 0.11 (0.03) | 0.13 (0.03) | 0.12 (0.06) | NS |

*Note*: Data are presented as mean (SD). Asterisks indicate sex differences within each exercise group.
*$P < 0.05$.
**$P < 0.01$.
***$P < 0.001$.
****$P < 0.0001$.
Height, $P = 0.0018$ (LOW F *vs*. M); $P = 0.0043$ (MOD F *vs*. M); $P = 0.00028$ (HIGH F *vs*. M). Body weight, $P = 0.0323$ (LOW F *vs*. M); $P = 0.0029$ (MOD F *vs*. M); $P = 0.0024$ (HIGH F *vs*. M). Lean mass, $P = 0.0001$ (LOW F *vs*. M); $P < 0.0001$ (both MOD F *vs*. M and HIGH F *vs*. M). VO$_2$peak (ml/min), $P = 0.002$ (HIGH F *vs*. M).
Abbreviations: BMI, body mass index; DBP, diastolic blood pressure; NS, not significant; RMR, resting metabolic rate; SBP, systolic blood pressure; VO$_2$, volume of oxygen consumption.

and returned to the pre-exercise level at 1 hPost (main effect of time, $P = 0.348$) (Fig. 2*J*).

When accounting for the exercise-induced reduction in plasma volume, exercise did not significantly alter plasma concentrations of IL1$\beta$, IL6, IFN$\gamma$ or TNF$\alpha$ (Fig. 3*A*), but there was a main effect of exercise on the increase in IL10 (main effect of time, $P = 0.0251$) (Fig. 3*A*). In contrast, without adjusting for plasma volume, we found a significant main effect of exercise on the concentrations of IL1$\beta$, IFN$\gamma$, TNF$\alpha$ and IL10 at End (main effect of time, $P < 0.001$ for IL1$\beta$, IFN$\gamma$ and TNF$\alpha$; $P = 0.0013$ for IL10) (Fig. 3*B*), except for IL6 (main effect of time, $P = 0.183$).

Collectively, these findings suggest that the increase in circulating cytokine bioavailability may be largely attributed to exercise-induced haemoconcentration.

## Acute LOW, MOD and HIGH induced distinct transcriptomic responses in aSAT

A total of 17,466 genes were detected by our RNA-seq analysis, and comparing aSAT samples collected before exercise (Pre) and 1.5 h after exercise (Post), we found 1003 differentially expressed genes (DEGs) after LOW (397 upregulated and 606 downregulated), 1979 DEGs

after MOD (773 upregulated and 1206 downregulated) and 759 DEGs after HIGH (452 upregulated and 307 downregulated) (adjusted $P < 0.05$) (Fig. 4$A$–$C$). Interestingly, there was a relatively small degree of overlap in DEGs detected among exercise groups (Fig. 4$D$), but principal component analysis (PCA) revealed distinct gene expression patterns ($\Delta$1.5 hPost–Pre), particularly in the HIGH group compared to MOD and LOW (Fig. 4$E$). Competitive gene set tests also identified distinct sets of biological pathways significantly altered by each of the three exercise treatments. For example, the top-upregulated pathways after LOW included protein translation and immune activation pathways, whereas pathways related to adipogenesis and lipid metabolism were downregulated (Fig. 4$F$). The upregulated pathways in MOD included protein translation, ribosome biogenesis and mitochondrial metabolism, whereas extracellular matrix (ECM) protein organization, $\beta$-catenin binding and thyroid hormone signalling were downregulated (Fig. 4$G$). HIGH robustly upregulated inflammation-related pathways, including complement, cytokine and FC receptor activities, IL6 signalling, IL10 production and angiogenesis, whereas pathways involved in protein translation, collagen assembly and mitochondria were downregulated (Fig. 4$H$).

Next we compared DEGs across our three different exercise treatments. We identified 734 DEGs in HIGH *vs.* MOD (larger changes in expression: 468 in HIGH and 266 in MOD) and 183 DEGs in HIGH *vs.* LOW (larger changes in expression: 153 in HIGH and 30 in LOW) (all adjusted $P < 0.05$) (Fig. 4$I$ and $J$). Surprisingly no DEGs were detected when comparing LOW *vs.* MOD (Fig. 4$K$). Protein secretion, glucose transport and insulin signalling pathways were enriched in HIGH compared to MOD and LOW (Fig. 4$L$ and $M$), whereas ribosome biogenesis and oxidative phosphorylation were more enriched in MOD compared to HIGH (Fig. 4$L$). Although no specific DEGs were found between LOW and MOD, gene set testing suggested that oxidative phosphorylation and

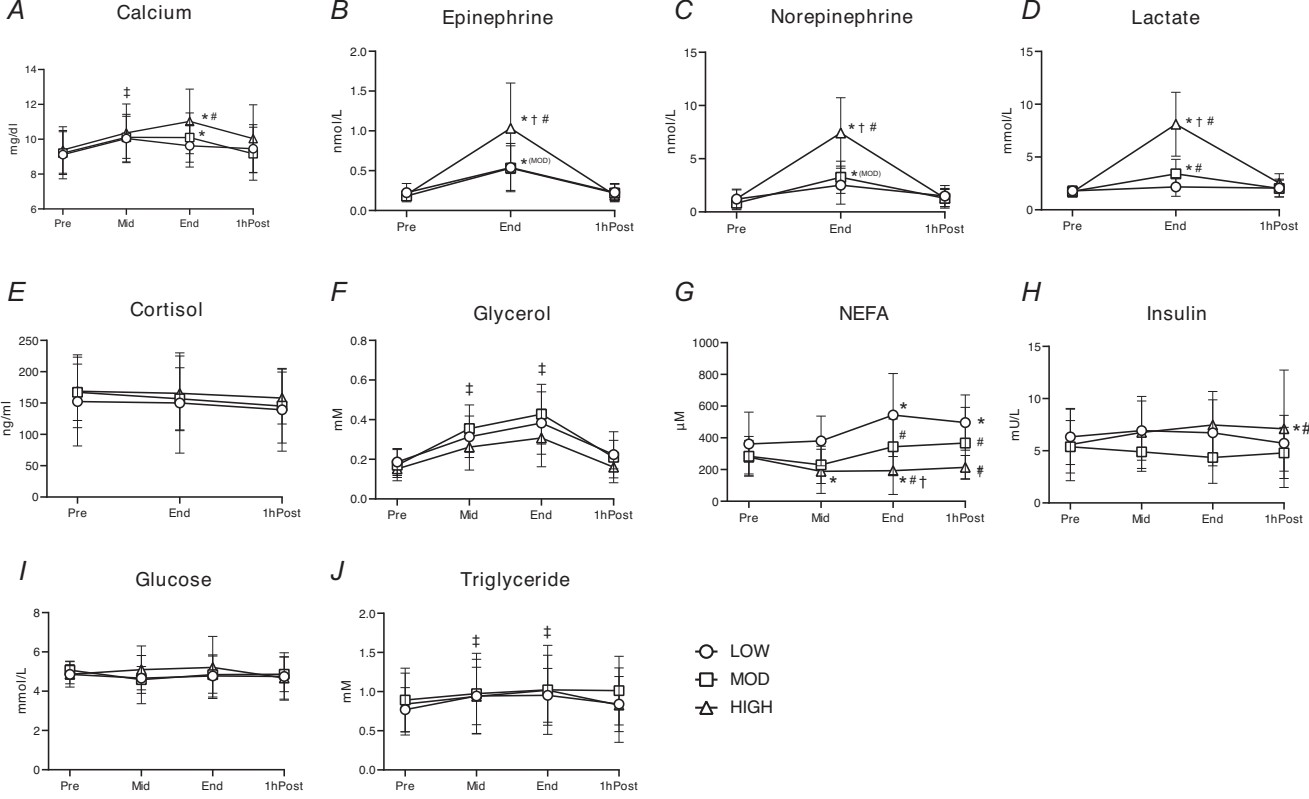

**Figure 2. Concentrations of circulating factors during and post LOW, MOD and HIGH**
Plasma concentrations of analytes were adjusted for blood volume using the following: *A*, calcium concentration. Blood volume-adjusted plasma concentration of *B*, epinephrine, *C*, norepinephrine, *D*, lactate, *E*, cortisol, *F*, glycerol, *G*, non-esterified fatty acids, *H*, insulin, *I*, glucose and *J*, triglyceride. ‡main effect of time ($P < 0.05$); *$P < 0.05$ *vs.* Pre; †$P < 0.05$ *vs.* MOD; #$P < 0.05$ *vs.* LOW. Significant overall time x group interaction effects were detected in epinephrine, norepinephrine and lactate. ($P < 0.001$, type III ANOVA). For calcium pairwise comparisons revealed significance between Pre and MID in all groups (LOW, $P = 0.032$; MOD, $P = 0.035$; HIGH, $P = 0.023$). There was a significant time (i.e. 1 hPost) x group (i.e. HIGH) interaction effect for insulin ($P = 0.025$). Data are presented as mean $\pm$ SD.

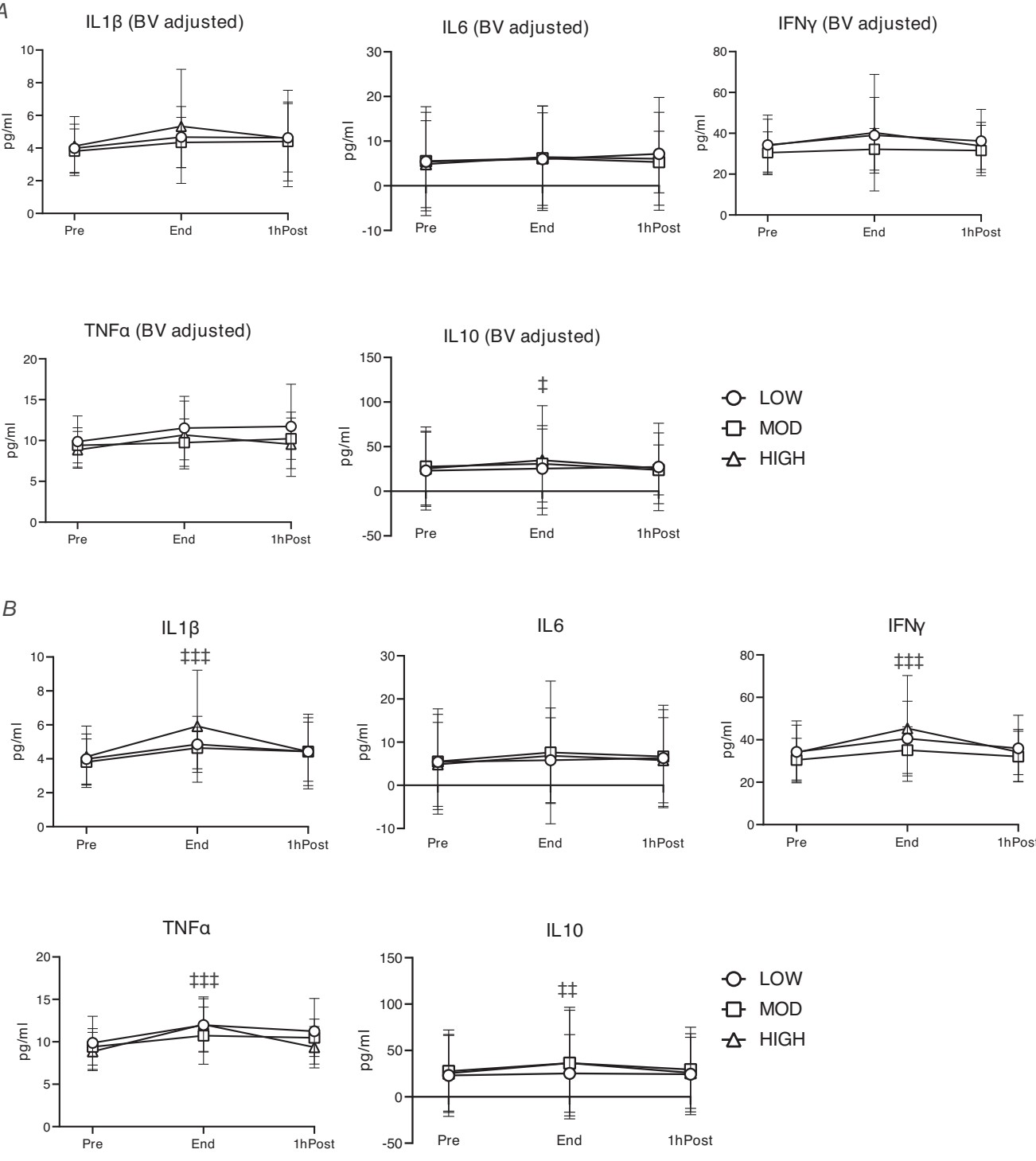

**Figure 3. Concentrations of circulating cytokines during and post LOW, MOD, and HIGH**
*A*, blood volume-adjusted plasma interleukin 1β (IL1β), IL6, interferon γ (IFNγ), tumour necrosis factor α (TNFα) and IL10 concentrations. *B*, unadjusted plasma IL1β, IL6, IFNγ, TNFα and IL10 concentrations. ‡Main effect of time ($P < 0.05$); ‡‡main effect of time ($P < 0.01$); ‡‡‡main effect of time ($P < 0.001$). Sample sizes, LOW: $n = 15$; MOD: $n = 15$; HIGH: $n = 15$. Data are presented as mean ± SD. BV, blood volume.

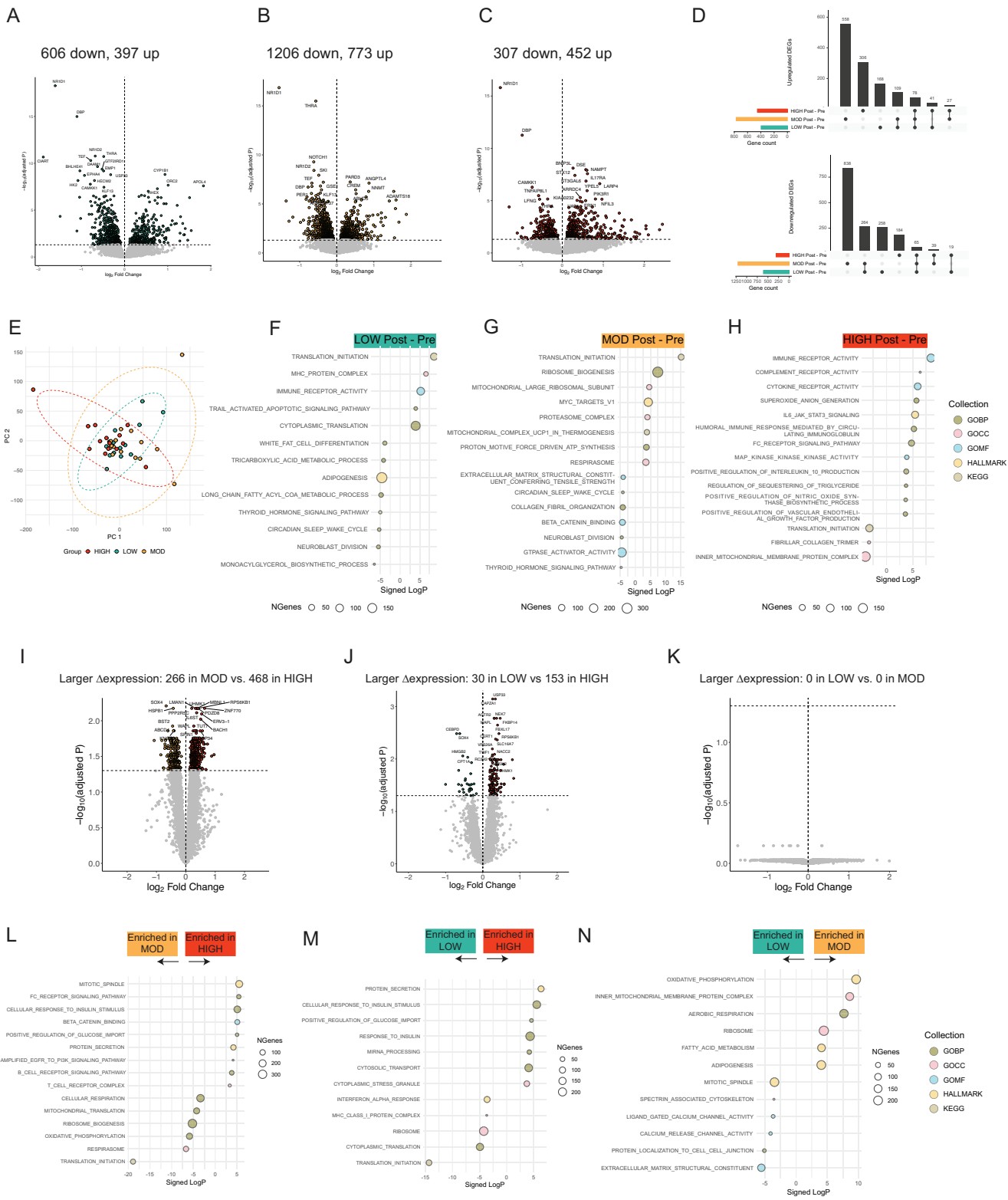

**Figure 4. Transcriptomic responses in abdominal subcutaneous adipose tissue (aSAT) 1.5 h after LOW, MOD and HIGH**

*A–C*, volcano plots of DEGs in LOW, MOD and HIGH 1 hPost *vs*. Pre (adjusted *P* < 0.05). *D*, upset plots of upregulated and downregulated DEGs after LOW, MOD and HIGH. Black dots indicate the corresponding bar plots of differentially expressed genes (DEG) count. Connecting lines indicate shared DEGs among exercise groups. *E*, principal component analysis (PCA) plot of changes in gene expressions (Δ1 h Post/Pre) for each exercise group.

*F–H*, enriched pathways after (*F*) LOW, (*G*) MOD and (*H*) HIGH (FDR < 0.05). Top 20 significant pathways (10 upregulated and 10 downregulated; adjusted *P* < 0.05) are shown. *I–K*, volcano plots of DEGs comparing the acute effects of (*I*) HIGH *vs*. MOD, (*J*) HIGH *vs*. LOW and (*K*) MOD *vs*. LOW (adjusted *P* < 0.05). Here DEGs indicate a significantly larger change in gene expression from Pre to 1 hPost in one group compared with another. *L–N*, enriched pathways comparing (*L*) HIGH *vs*. MOD, (*M*) HIGH *vs*. LOW and (N) MOD *vs*. LOW (FDR < 0.05). Top 20 significant pathways (10 upregulated and 10 downregulated; FDR<0.05) are shown. GOBP, Gene Ontology – Biological Process; GOCC, Gene Ontology – Cellular Component; GOMF, Gene Ontology – Molecular Function; KEGG, Kyoto Encyclopedia of Genes and Genomes.

mitochondrial respiration pathways were more enriched in MOD compared to LOW (Fig. 4*N*).

## Circadian clock genes are strongly altered by acute exercise

Although the overlap of DEGs and enrichment of biological pathways among the three exercise groups was less than anticipated, closer examination of the DEGs that were altered the most (i.e. $|\log_2 FC|>1$) revealed significant changes in many circadian clock genes across all treatments. Notably a core clock gene nuclear receptor subfamily 1 group D member 1 (*NR1D1*; also referred to as Rev-Erba), which inhibits the transcription of master clock gene *BMAL1*, was the most significantly downregulated gene in all exercise groups (Fig. 5*A*). Other core clock genes, such as circadian-associated repressor of transcription (*CIART*), D-box binding PAR bZIP transcription factor (*DBP*) and period circadian regulator 1 (*PER1*), were also suppressed by exercise. Conversely nicotinamide phosphoribosyltransferase (*NAMPT*), also known to regulate circadian rhythm (Ramsey et al., 2009), was upregulated by all treatments

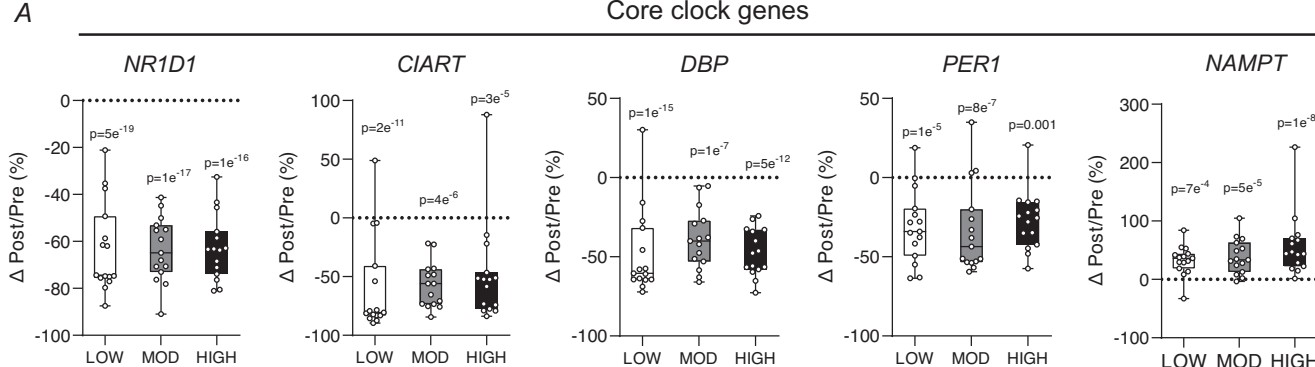

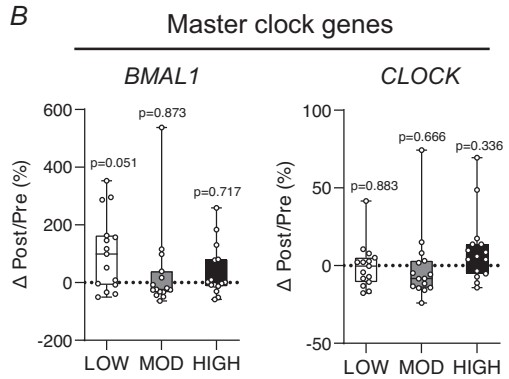

**Figure 5. Alterations in circadian clock genes in abdominal subcutaneous adipose tissue (aSAT) 1.5 h after exercise**
*A*, percentage change of core clock genes (Δ1 hPost/Pre) after LOW, MOD and HIGH. *B*, percentage change of master clock genes (Δ1 hPost/Pre) after LOW, MOD and HIGH. *P*-values represent the BH-corrected adjusted *P*-value obtained from DESeq2 analysis.

in most subjects (Fig. 5*A*). We did not observe any exercise-induced changes in the gene expression of *BMAL1* or *CLOCK* (Fig. 5*B*), which are considered the 'master clock' transcription factors that regulate the expression of other circadian genes (Gekakis et al., 1998).

## Gene modules identified by network analysis are linked to clinical traits

We conducted WGCNA on pre-exercise transcripts to identify gene clusters (i.e. modules) and their ME (first principal component of module that represents overall gene expression of the module) that may help identify how genes are co-expressed, and how they may be functionally related. A total of 16 modules were identified (including the 'grey' module – cluster of non-correlating genes, Module 0), each containing at least 200 genes. After module construction we performed overrepresentation analysis to assess enriched biological pathways for each module (Fig. 6*A*), associated ME with clinical/sub-clinical traits (Fig. 6*B*) and also mapped acute exercise effects by quantifying the overlap of module member genes with modality-specific DEGs (Fig. 6*C*). For sub-clinical traits we measured fat cell size, overall collagen deposition (i.e. Sirius Red) and adipose tissue macrophages (CD14 for pro-inflammatory and CD206 for anti-inflammatory) (Fig. 6*D*). ME1 was enriched in pathways related to translation, aerobic respiration and lipid metabolic process (Fig. 6*A*), and its 'hub genes' (i.e. highly connected genes within ME) included mitochondrial genes (e.g. *SDHB, NDUF, COX*) (Fig. 6*E*). ME1 was linked to females ($P < 0.001$), inversely associated with waist circumference ($P < 0.001$) and fat cell size ($P = 0.0287$) (Fig. 6*B*), and a distinct number of its member genes were differentially regulated by LOW ($n = 121$), MOD ($n = 228$) and HIGH ($n = 65$) (Fig. 6*C*). ME8 was enriched for inflammation, cell migration and lipid metabolism (Fig. 6*A*), with its top 10 hub genes being key factors in immune cell activation, migration and cytokine production (Fig. 6*E*). Additionally ME8 was positively associated with adiposity (fat mass, $P = 0.006$; BMI, $P = 0.010$; waist circumference, $P < 0.001$), fasting glucose concentration ($P = 0.032$), fat cell size ($P < 0.001$) with 26, 41 and 52 genes being differentially regulated by LOW, MOD and HIGH, respectively (Fig. 6*C*).

Then we examined whether certain modules may be more responsive to acute exercise. Notably ME9 was enriched for pathways involved in translation and RNA processing (Fig. 6*A*), and its ME was significantly downregulated by MOD ($P = 0.048$, Fig. 6*F*). Strikingly nearly 25% of ME9 genes were altered by MOD compared to ∼3% in LOW and HIGH (Fig. 6*C*). Conversely we found that ME9 was increased in HIGH ($P = 0.041$, Fig. 6*F*). Several other MEs were also impacted by acute exercise:

ME3 increased after LOW ($P = 0.011$), ME12 increased after MOD ($P = 0.021$) and ME3 and ME4 decreased after HIGH ($P = 0.023$ and $P = 0.022$, respectively) (Fig. 6*F*).

To explore potential transcriptional regulation in aSAT by circulating factors altered during exercise, we correlated the change in unadjusted blood analytes from Pre to End ($\Delta$) with changes in ME from Pre to 1.5 hPost ($\Delta$ME). $\Delta$Insulin, known for its role in inhibiting adipocyte lipolysis, was negatively associated with $\Delta$ME13 ($P = 0.023$) and $\Delta$ME14 ($P = 0.016$) (Fig. 6*G*). Notably $\Delta$ME3 showed a positive correlation with $\Delta$norepinephrine ($P = 0.030$) but an inverse relationship with several circulating cytokines such as $\Delta$IL10 ($P < 0.001$), $\Delta$IL1$\beta$ ($P = 0.004$), $\Delta$IL6 ($P = 0.021$) and $\Delta$TNF$\alpha$ ($P = 0.023$), suggesting that genes in ME3 may be transcriptionally regulated by exercise-induced hormonal changes (Fig. 6*G*). Among the top 20 hub genes for ME3, *ANXA7, ATL3, CCPG1, DDR2, EPDR1, GNS, LPGAT1, RNF141, SEMA3C, VLDLR* and *YAP1* were differentially regulated by at least one exercise modality (adjusted $P < 0.05$, Fig. 6*H*). These genes span functions in ER and membrane trafficking (Gerke & Moss, 1997; Smith & Wilkinson, 2018), as well as mechanosensation (Liu et al., 2023), pointing to their potential roles in early transcriptional response to circulating factors in aSAT. Interestingly, some of these genes (i.e. *GNS, LNPEP, SEMA3C*) have been reported to encode secretory proteins from human adipose tissue (Ahn, Tamburi Ni, et al., 2025; Mejhert et al., 2013), suggesting a link between exercise-induced cytokine response and adipokine production in aSAT, which may be dependent on exercise intensity (Fig. 6*H*).

## Acute exercise downregulated ERK phosphorylation at 1.5 hPost

Targeted analysis revealed that exercise decreased the phosphorylation of ERK in aSAT (expressed as the ratio of protein abundance of pERK$^{T202/Y204}$:total ERK) after exercise (main effect of time, $P = 0.047$), with no differences observed among groups (Fig. 7*A*). In contrast, we did not detect an effect of exercise on the phosphorylation of P38 (ratio pP38$^{T180/Y182}$:total P38) or STAT3 (ratio pSTAT3$^{Y705}$:total STAT3) in any of the groups (Fig. 7*B* and *C*). Phosphorylation of one of the chief lipolytic enzymes, HSL, at either the AMPK regulatory site serine 565 (expressed as pHSL$^{S565}$:total HSL) or the PKA regulatory site, serine 660 (pHSL$^{S660}$:HSL), was not increased in aSAT collected 1.5 h after any of the exercise sessions (Fig. 7*D*). This aligns with the plasma epinephrine concentration, which returned to pre-exercise levels 1 h after exercise in all groups (Fig. 2*B*). Similarly, there were no differences in the phosphorylation

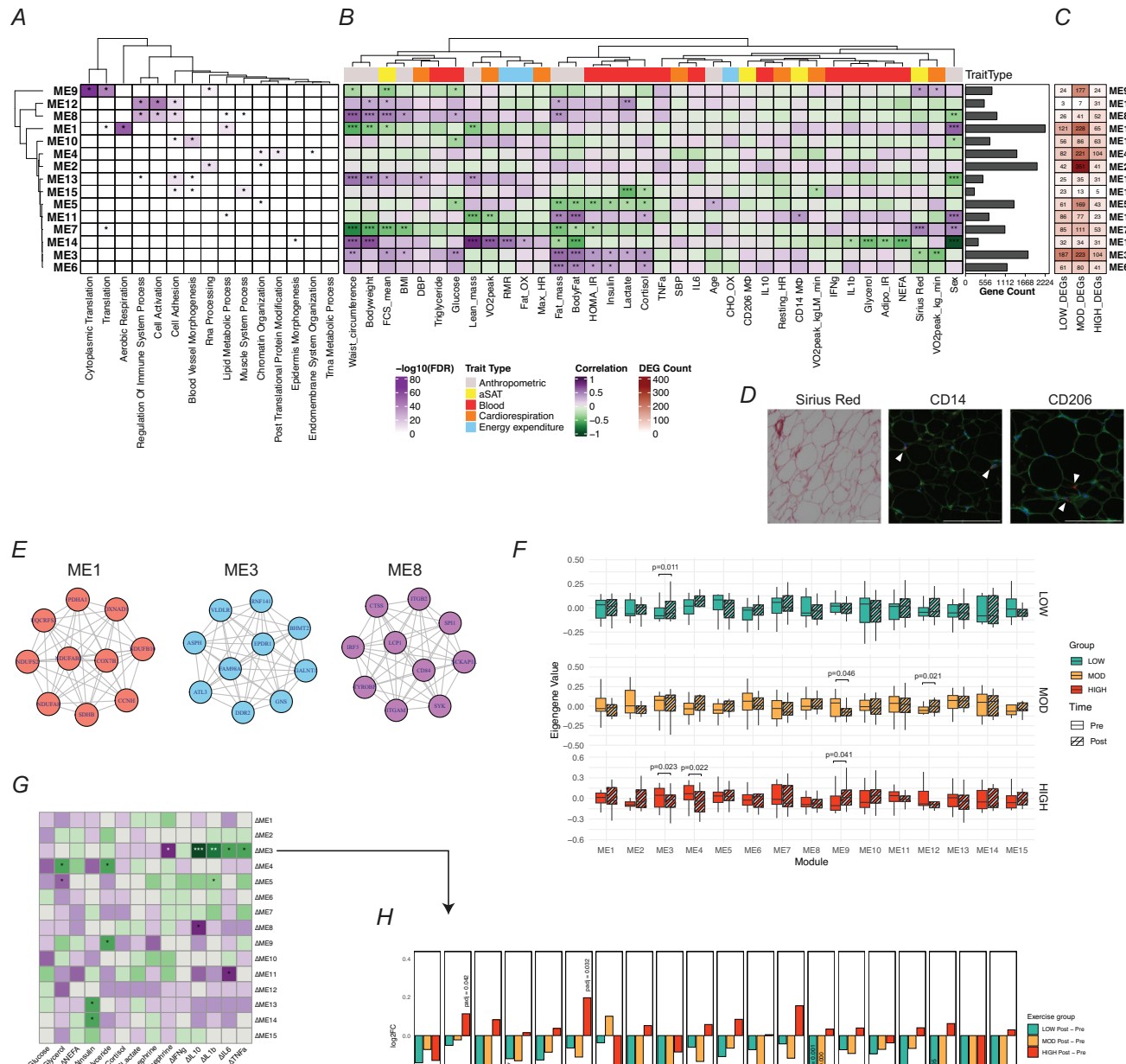

**Figure 6. Integration of transcriptomics with clinical/tissue traits**

Weighted gene co-expression network analysis (WGCNA) was performed on the pre-exercise transcriptomics data (*n* = 45). *A*, overrepresentation analysis (ORA) was conducted on module member genes. Significantly over-represented terms are marked with asterisk (false discovery rate (FDR) < 0.05). Colour represents −log$_{10}$(FDR). *B*, module eigengene (ME) was correlated with clinical and sub-clinical traits using biweight midcorrelation. *P < 0.05; **P < 0.01; ***P < 0.001. Module size (number of genes in each module) is shown in horizontal bar graphs, right to the correlation heatmap. *C*, differentially expressed genes (DEGs) were mapped to module member

genes. Count of DEGs is shown in numbers. *D*, representative images of abdominal subcutaneous adipose tissue (aSAT) histology. Sirius Red was used to stain collagen type I and III depositions. CD14 was used as a marker for pro-inflammatory macrophages. CD206 was used as a marker for anti-inflammatory macrophages. Positive stains of CD14 and CD206 are marked with white arrows. White bars indicate 100 μm. *E*, top 10 hub genes from selected modules. Grey edges indicate intercorrelation between genes. *F*, change in ME after LOW, MOD and HIGH. *G*, correlation of ΔME (1.5 hPost/Pre) with Δblood analytes (1 hPost/Pre). *P < 0.05; **P <0.01; ***P < 0.001. Bicor, biweight midcorrelation r value. *H*, acute exercise responses in top 20 hub genes of ME3. Bar plots in the first row represent exercise modality-specific effect (Post–Pre). Plots in the second row represent a direct comparison of exercise modalities. Adjusted *P*-values are shown only for DEGs.

of AKT (pAKT$^{T308}$:AKT or pAKT$^{S473}$:AKT) in the aSAT samples collected after LOW, MOD or HIGH (Fig. 7*E*).

### Integration with single-cell data suggests cell-type dependent responses in aSAT by acute exercise

Using cell-type markers acquired from three previously published reports (Backdahl et al., 2021; Emont et al., 2022; Whytock et al., 2024), we found a strikingly distinct pattern of transcriptional activation among cell types that were differentially regulated by the three different exercise treatments (Fig. 8*A*). Notably, marker genes for infiltrating mesenchymal immune cells (i.e. macrophage, monocyte, neutrophil) were upregulated in HIGH from all three cell-type marker gene sets (Fig. 8*A*), suggesting that these cells may underlie the strong upregulation of inflammatory pathways observed in HIGH (Fig. 4*H*). Conversely, marker genes for adipocytes were downregulated in LOW (Fig. 8*A*), consistent with the acute downregulation of 'Adipogenesis' pathway we found in our RNA-seq analysis (Fig. 4*F*). Additionally, we observed a potential exercise intensity-dependent effect on vascular cells, with LOW upregulating gene expression in endothelial and vascular cells, whereas HIGH exerted the opposite effect (Fig. 8*A*).

Building on the aSAT single-nuclei 'signature matrix' from 10 individuals (age 18–40) reported from Whytock et al. we applied a recently developed deconvolution algorithm optimized for human aSAT (Ahn, Divoux, et al., 2025, Whytock et al., 2024) (see details in the Methods) to estimate the proportion (%) of each cell type before and after acute exercise. Deconvolution analysis revealed an increased estimated proportion of macrophages in aSAT 1.5 hPost (main effect of time; *P* = 0.010), suggesting the possibility of rapid infiltration of immune cells into aSAT after exercise (Fig. 8*B*). In contrast, the estimated proportion of adipocyte 1, a sub-type of adipocytes that is characterized by high expression of genes involved in anti-oxidant activity (*GPX1* & *GPX4*) (Whytock et al., 2024), was reduced 1.5 hPost (main effect of time; *P* = 0.011). The estimated proportion of adipocyte 2, enriched with upregulation of insulin signalling pathways (Whytock et al., 2024), was only reduced by LOW (Fig. 8*B*). Additionally the estimated proportion of vascular cells was reduced only by HIGH (*P* = 0.016)

(Fig. 8*B*). However it is important to note that reductions in relative proportions may be driven by the increased proportion of macrophages. Overall our integration of single-cell data suggests that early transcriptional and cellular responses in aSAT may be differentially regulated after a session of exercise performed at different intensity and energy expenditure.

## Discussion

The tissue and cellular adaptations that occur with chronic exercise training result from the accumulation of signalling events/responses to each session of exercise (Egan & Zierath, 2013). In this study we aimed to discover the early transcriptomic responses in aSAT triggered after a session of exercise that may contribute to the maintenance or remodelling of aSAT milieu. Among the most novel findings of this study were some distinct alterations in aSAT transcriptome observed 1.5 h after the three different exercise sessions (LOW, MOD, HIGH), all of which are commonly prescribed for maintaining fitness and health. HIGH induced the most pronounced increase in inflammatory and angiogenic genes, whereas MOD and LOW increased genes regulating ribosomal and oxidative phosphorylation pathways. Despite a relatively small degree overlap of DEGs among the three groups, alteration in circadian clock genes was largely consistent across three exercise modalities accompanied by reduced phosphorylation of ERK protein and altered circulating cytokine abundances. Our comprehensive bioinformatic analyses provide insights into the clinical relevance of exercise-induced alterations in aSAT transcriptomes and cell types. These molecular modifications in aSAT that occur shortly after each session of exercise may contribute to the maintenance of adipose tissue function in those who exercise regularly.

We recently reported that exercise training induced some structural and proteomic adaptations in aSAT that were independent of changes in fat mass, and these adaptations were linked to improved cardiometabolic health profiles (Ahn et al., 2022, 2024). These adaptations included increased vasculature, altered ECM collagen profile and elevated abundances of ribosomal, mitochondrial and lipogenic proteins in aSAT (Ahn et al., 2022, 2024). In the present study,

our assessments of changes in transcriptional pathways that occur shortly after a session of exercise – including rapid alterations in pathways regulating vasculature development, ECM remodelling, ribosome biogenesis, oxidative phosphorylation and lipid metabolism – provide valuable insights into the acute effects of exercise on aSAT that may underlie long-term adaptations. Notably, many of these responses to acute exercise were different among our three exercise treatments that differed in intensity and energy expenditure. For example, genes for factors involved in adipose tissue ECM protein organization were increased in MOD and HIGH, but an increase in genes involved in the regulation of angiogenesis was observed only after HIGH. Similarly pathways regulating

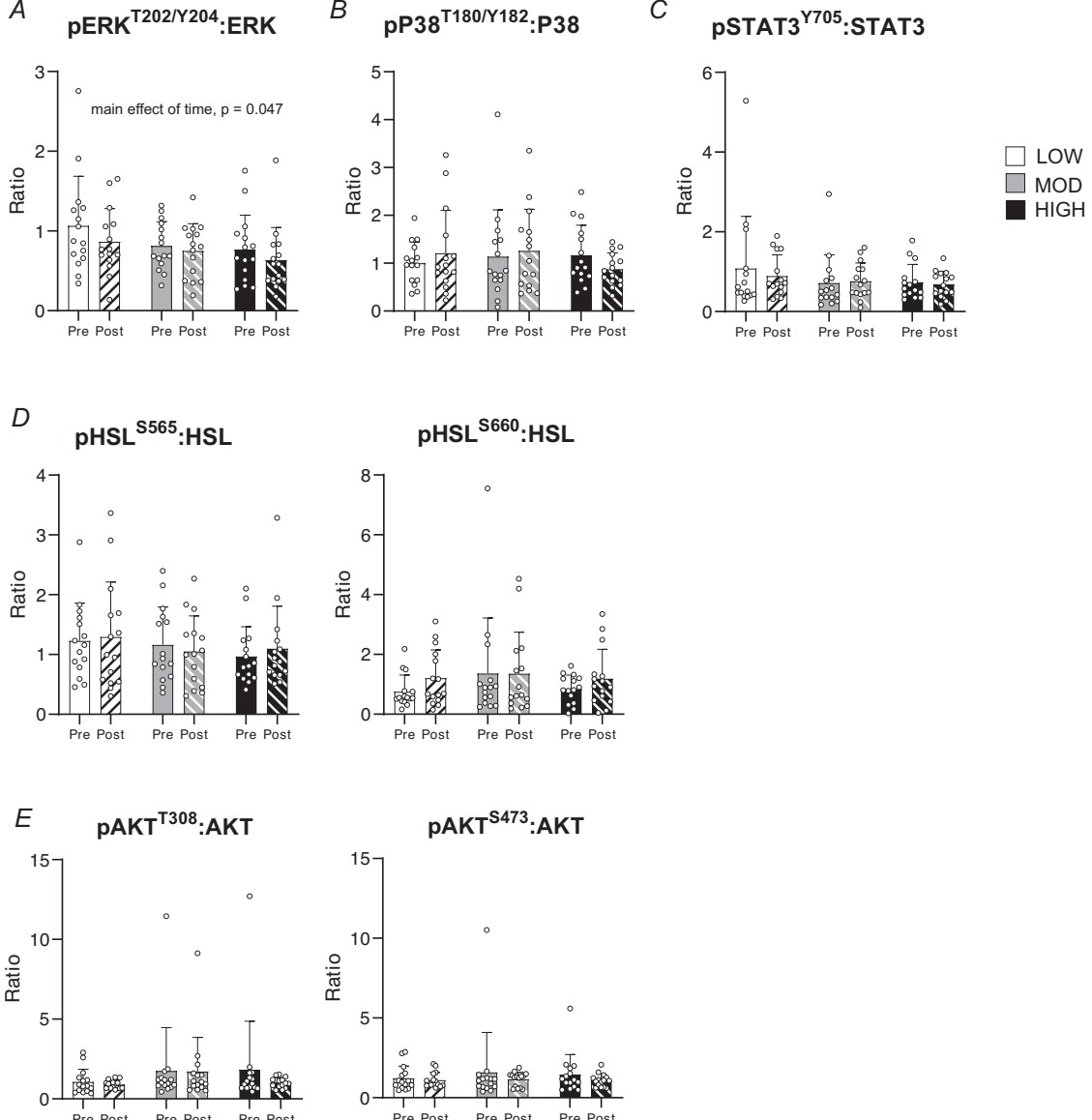

**Figure 7. Protein phosphorylation of metabolic proteins in abdominal subcutaneous adipose tissue (aSAT) 1.5 h after LOW, MOD and HIGH**

*A*, ratio of pERK$^{T202/Y204}$:ERK (main effect of time, $P = 0.047$; group, $P = 0.148$; time*group, $P = 0.928$). *B*, ratio of pP38$^{T180/Y182}$:P38 (main effect of time, $P = 0.311$; group, $P = 0.605$; time*group, $P = 0.614$). *C*, ratio of pSTAT3$^{Y705}$:STAT3 (main effect of time, $P = 0.782$; group, $P = 0.754$; time*group, $P = 0.910$). *D*, ratio of pHSL$^{S565}$:HSL (main effect of time, $P = 0.725$; group, $P = 0.699$; time*group, $P = 0.573$) and pHSL$^{S660}$:HSL (main effect of time, $P = 0.210$; group, $P = 0.798$; time*group, $P = 0.668$). *E*, ratio of pAKT$^{T308}$:AKT (main effect of time, $P = 0.607$; group, $P = 0.293$; time*group, $P = 0.588$) and pAKT$^{S473}$:AKT (main effect of time, $P = 0.636$; group, $P = 0.951$; time*group, $P = 0.711$). *$P < 0.05$ *vs.* Pre. There was no significant time × group interaction effect. Data are presented as mean ± SD.

processes such as protein secretion, insulin signalling and glucose uptake were uniquely enriched by HIGH but not in MOD or LOW. These findings suggest that a relatively high exercise-intensity threshold may be required to stimulate gene transcription for factors regulating ECM remodelling, vascularization, endocrine function and insulin action in aSAT. Conversely, oxidative phosphorylation and ribosome biogenesis were more enriched by MOD compared to LOW or HIGH. Because the estimated energy expended during MOD (~400 kcal) was greater than both LOW and HIGH (both ~250 kcal),

it is possible that these responses might be sensitive to factors related to substrate utilization.

Expression of inflammatory genes has been reported to increase in adipose tissue after a session of exercise in both preclinical models (Castellani et al., 2015; Macpherson et al., 2015) and in humans (Fabre et al., 2018; Keller et al., 2003; Ludzki et al., 2021), suggesting that a transient induction of aSAT inflammation may be an intrinsic response to the stress of exercise. Factors responsible for this exercise-induced inflammatory response in aSAT are not completely understood, but an accumulation of

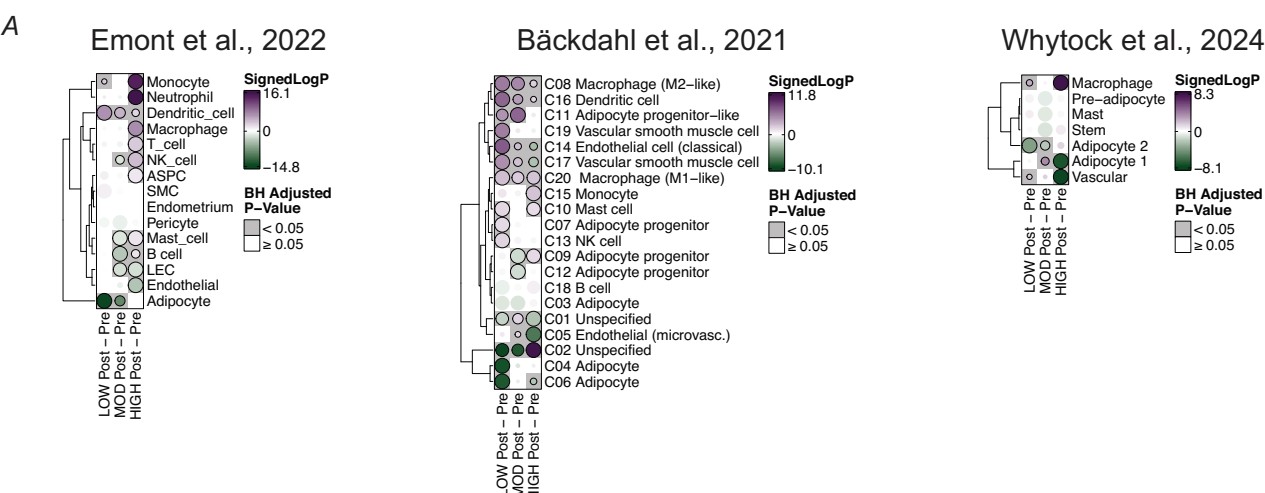

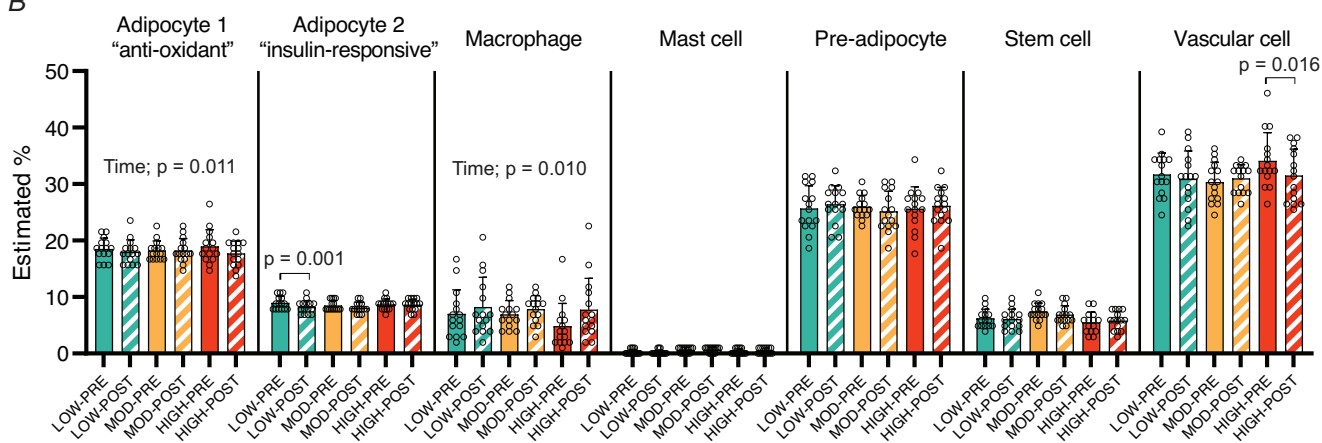

**Figure 8. Cell-type specific responses and estimated cell-type proportion of abdominal subcutaneous adipose tissue (aSAT) 1.5 h after LOW, MOD and HIGH**

*A*, cell-type enrichment analysis using cell-type markers derived from three publicly available single-cell RNA-sequencing datasets (Emont et al. and Whytock et al.) or spatial transcriptomics (Backdahl et al.). Competitive gene testing was used to assess cell-type enrichment by exercise. The size of each circle represents the absolute signed $\log_{10}$(adjusted *P*-value). Positive signed $\log_{10}$(adjusted *P*-value) represents activation, and negative signed $\log_{10}$(adjusted *P*-value) represents suppression of cell-type gene markers. *B*, estimated proportion of cell types via deconvolution analysis using signature matrix from Whytock et al. (2024). Time, main effect of time. ASPC, adipose stem and progenitor cells; LEC, lymphatic endothelial cells; NK, natural killer; SMC, smooth muscle cells.

non-esterified fatty acids may be an important contributor (Hodgetts et al., 1991; Tsiloulis & Watt, 2015). The robust upregulation of genes involved in inflammatory signalling and immune cell activations (e.g. macrophages, B cell, T cell) that we observed in HIGH may have been, at least partly, due to the restriction in adipose tissue blood flow known to occur during high-intensity exercise (Nguyen et al., 2007; Yu et al., 2002), and the resultant 'trapping' of non-esterified fatty acids within the tissue (Hodgetts et al., 1991; Tsiloulis & Watt, 2015), reflected in our study by the low plasma fatty acid concentration during HIGH. Similarly, a relatively high rate of lipolysis relative to energy expenditure during low-intensity exercise (Romijn et al., 1993) may result in an intracellular accumulation of non-esterified fatty acids, which, in turn, may contribute to the upregulation in aSAT inflammation we observed in LOW. Our finding that inflammatory pathways were not upregulated in MOD contrasts with some previous studies (Keller et al., 2003; Ludzki et al., 2021), but these earlier studies were conducted in untrained subjects. Because exercise training attenuates the inflammatory response in adipose tissue after a session of moderate-intensity exercise (Fabre et al., 2018), the lack of an inflammatory response after MOD in our regular exercisers was not unexpected. Collectively, these findings suggest that the upregulation of gene expression in immune and inflammatory pathways in aSAT during acute exercise may be dependent on both exercise intensity and the individual's training history.

Our novel findings also demonstrate that the expression of genes for several core clock components (e.g. *NR1D1, NAMPT, DBP, PER1, CIART*) (Cox & Takahashi, 2019; Goriki et al., 2014) were significantly altered by acute exercise. It is well documented that lipid mobilization and storage in aSAT are impacted by circadian regulation (Paschos et al., 2012; Shostak et al., 2013), and the disruption of the circadian rhythm of these processes has been found to be linked to metabolic health complications (Bray & Young, 2007). Interestingly, the gene we found to have the greatest magnitude of change in response to all three exercise treatments was the reduction in gene expression of the core clock component, *NR1D1* (Rev-Erbα), which was recently reported to play a critical role in adipose tissue abnormalities common in obesity (e.g. fibrotic ECM and macrophage infiltration) (Hunter et al., 2021). Hunter et al. demonstrated that compared to wild-type mice on a high-fat diet (HFD) mice with a specific deletion of *NR1D1* in adipocytes were protected from the large increase in fibrosis and macrophage infiltration in adipose tissue in response to the same HFD despite an even greater increase in adiposity in the knockout animals (Hunter et al., 2021). The adipocyte-specific *NR1D1* knockout mice were also protected from developing insulin resistance in response to the HFD (Hunter et al., 2021). Additionally the

increased gene expression of *NAMPT* that we observed after exercise also supports the notion that acute exercise may trigger cues to favourably modify aSAT because *NAMPT*, a key enzyme that mediates NAD+ biosynthesis and circadian lipid metabolism (Garten et al., 2015), has been found to be important in the healthy expansion of subcutaneous adipose tissue (Nielsen et al., 2018). Because a greater *capacity* to store lipid in subcutaneous adipose tissue can reduce excessive ectopic lipid deposition in tissues such as the liver, skeletal muscle, pancreas, etc., thereby improving cardiometabolic health (Cypess, 2022; Ravussin & Smith, 2002), it is intriguing to consider that exercise-induced changes in the expression of *NR1D1* and *NAMPT* in aSAT after each session of exercise may lead to beneficial adaptations by reducing ectopic lipid deposition and the resultant dysfunction in metabolically active tissues.

The reduction in *PER1* we observed after exercise is in contrast with a recent finding that a session of moderate-to-vigorous exercise (80% $VO_2$max) increased the mRNA expression of *PER1* in aSAT after 8 weeks of exercise training in adults with overweight/obesity (Dreher et al., 2023). Reasons for this discrepancy are not clear, but this suggests changes in circadian transcripts after acute exercise may depend on factors such as the degree of adiposity, training and/or nutritional status (Dreher et al., 2023; Strączkowski et al., 2023). Interestingly the expression of 'master clock' genes (i.e. *BMAL1* and *CLOCK*), whose transcription is suppressed by *NR1D1* and *PER* (Dollet & Zierath, 2019), did not change after acute exercise in either our study or that of Dreher et al. (Dreher et al., 2023), which may indicate that the main aSAT clock was stable. We acknowledge that it is possible we may not have captured an exercise-induced change in expression of the master clock genes in our samples collected 1.5 h after the exercise session. In addition, although our finding of exercise-induced alterations in core clock genes is interesting, because circadian rhythm is tightly associated with energy homeostasis in aSAT (Asher & Sassone-Corsi, 2015), we acknowledge that these responses may be driven by an indirect effect of the energy deficit induced by exercise or circadian rhythmicity, rather than a direct effect of the exercise, per se.

Our findings that plasma concentrations of circulating pro- and anti-inflammatory cytokines were elevated at the end of moderate- and high-intensity exercise align with previous studies (Christiansen et al., 2013; Lira et al., 2017; Sim et al., 2013). Here we report that low-intensity exercise (40% $VO_2$max) was also sufficient to increase circulating concentration of cytokines. Therefore despite very different exercise stimuli and energy expenditure during LOW, MOD and HIGH in our study, the effect of exercise on the circulating cytokines we measured were remarkably similar. However, our findings suggest

that the increased concentration of circulating cytokines in response to acute exercise may be largely driven by exercise-induced haemoconcentration, rather than de novo secretion, except for IL10, which was the only cytokine we measured that remained significantly elevated after correction for the exercise-induced reduction in plasma volume. It is possible that an increased secretion of IL10 from aSAT may have contributed to the elevation in circulating IL10, particularly by high-intensity exercise, given the findings from our RNA-seq analysis indicating that cytokine production and IL10 biogenesis pathways were upregulated in HIGH.

We used WGCNA to identify highly correlated gene clusters, which enabled us to probe the possible impact of exercise-induced changes in aSAT transcripts on cardiometabolic health. For instance, in aSAT samples collected before exercise, the positive association between fat mass or adipocyte size and ME8 and ME12, modules enriched for inflammatory pathways, corroborates the well-established link between aSAT inflammation and adiposity (Weisberg et al., 2003). By integrating WGCNA modules and postexercise aSAT transcriptomics, we aimed to comprehensively characterize exercise-induced transcriptional responses in aSAT. Among the gene modules that were significantly altered by exercise the change in ME3, a module that was associated with 'unfavourable' cardiometabolic traits (e.g. higher adiposity, adipocyte size and fasting glucose, insulin and cortisol), was inversely correlated with changes in circulating cytokines such as IL10, IL1$\beta$, IL6 and TNF$\alpha$. Perhaps the increased concentrations of these cytokines during exercise may have acted to regulate the expression of ME3 genes. Collectively, this analysis bridges exercise-induced effects in aSAT with cardiometabolic health traits, providing clinical insights into the role of exercise on transcriptional regulation in aSAT.

ERK and P38 are key signalling proteins in MAPK pathway that are involved in the regulation of adipocyte proliferation, lipolytic function and inflammation in adipose tissue (Greenberg et al., 2001; Hu et al., 1996; Trujillo et al., 2006), and they have been found to be modified by exercise training (Stephenson et al., 2013; Trevellin et al., 2014; Zhang et al., 2014). The reduction in ERK phosphorylation we found after all three acute exercise sessions in our study suggests that a session of exercise may rapidly trigger signalling pathways that regulate the growth/differentiation of adipocytes (Bost et al., 2005). Because it is likely that ERK phosphorylation is essential for the early proliferative stages of adipogenesis in progenitor cells, but has to be shut off for preadipocytes to continue to differentiate (Hu et al., 1996; Sale et al., 1995), we speculate that the exercise-induced reduction in ERK activity may reflect cell-type–specific proliferative responses in progenitor cells *vs.* committed preadipocytes. However, this remains to be tested. In contrast, we did

not observe a difference in the phosphorylation of key lipolytic enzyme HSL or insulin signalling protein AKT, which aligns with our observation that the concentration of epinephrine, a lipolytic stimulator and insulin, an upstream activator of AKT returned to pre-exercise levels 1.5 h after exercise. Interestingly, despite the distinct differences in exercise intensity, duration and energy expenditure among groups the effect of acute exercise on the phosphorylation of the key metabolic proteins that we measured (ERK, P38, STAT3, HSL, AKT) – at least 1.5 h postexercise – was remarkably similar.

Our deconvolution analysis, which estimated cell-type proportions (%), revealed a postexercise increase in macrophages and a reduction in adipocytes and vascular cells, expanding our understanding of rapid cell-type–specific alterations in aSAT in response to acute exercise. Although transcriptional alterations, rather than changes in cell abundance, could explain the observed shifts in cell-type proportions, our previous study found a reduction in a specific preadipocyte subgroup 12 h after a 60 min session of endurance exercise in individuals with obesity, suggesting that acute exercise may influence cell-type abundances rather quickly (Ludzki et al., 2020). Our finding of an increased proportion of macrophages after acute exercise aligns with a previous preclinical study that reported an increased abundance of anti-inflammatory macrophages in white adipose tissue 4 h after acute swimming exercise in rats (Oliveira et al., 2013). Although our data suggest that acute exercise may rapidly recruit macrophages (and cause slight reduction in relative proportion of other cell types) and activate various immune cell types, it is important to note that inflammatory pathway upregulation in aSAT is largely short-lived (Ahn et al., 2022), and long-term exercise training may even reduce the overall abundance of macrophages in aSAT (Ahn et al., 2024). Although chronically elevated inflammation in adipose tissue and systemic circulation is commonly linked with a host of chronic diseases (Jia et al., 2020; Kawai et al., 2021; Suganami et al., 2012), paradoxically acute inflammatory responses or immune cell activations in aSAT may be important contributors to favourable tissue remodelling (Asterholm et al., 2014). Whether aSAT cell-type proportions can be altered by exercise training is still unclear. However our findings, along with previous findings from our laboratory (Ludzki et al., 2020), suggest that a session of endurance exercise may induce cell-type–specific responses that, when accumulated over time (i.e. training), may contribute to chronic adaptations in aSAT cellular proportions, as seen in mouse models (Nigro et al., 2023).

Despite considerable efforts to control for various aspects of our study, some limitations remain. Although participants served as their own control within each exercise group (i.e. pre/postexercise), the comparison

across the three different exercise sessions was not conducted using a repeated measures design. Importantly, we were successful in tightly matching subjects in each of the three exercise groups for key factors that could impact the outcomes, such as sex, cardiorespiratory fitness (VO$_2$peak), body fat mass and lean mass, enhancing confidence in the interpretations of our findings. Additionally, because we did not include a control trial in which adipose tissue samples were collected at the same time points without exercise, we cannot fully exclude the possibility that some of the observed exercise-induced responses in aSAT were influenced by factors unrelated to exercise, such as fasting duration or circadian rhythms. Furthermore, although our three different exercise treatments (LOW, MOD and HIGH) represent commonly prescribed exercise programmes, the three exercise sessions were not matched for exercise duration or energy expenditure, which may confound our interpretation. However, the goal of our study was to compare three distinct exercise modalities that are commonly implemented in clinical/applied settings, which makes matching for exercise duration and/or energy expenditure unfeasible. It is also important to acknowledge that the responses observed in the HIGH condition cannot be solely attributed to the high-intensity intervals themselves. The non-steady-state nature of this exercise may have influenced the observed adaptations as well. However, exercise intensity used during HIGH (which mimics common high-intensity interval training (HIIT) prescriptions) cannot be performed continuously for more than a few minutes at most. Another important limitation is that collecting only one aSAT sample ∼1.5 h after exercise obviously does not fully capture the acute effect of exercise on aSAT that can persist for at least several hours or even a few days. Therefore our findings are limited to characterizing the very early responses to exercise performed at low-, moderate- and high-intensity exercise.

In summary, our findings indicate that acute exercise can rapidly modify aSAT transcriptome, and some of these modifications were markedly different after the LOW, MOD and HIGH exercise sessions in our study. Specifically angiogenesis, protein secretion and insulin signalling pathways were enriched by HIGH, whereas ECM remodelling, ribosome biogenesis and oxidative phosphorylation were induced by MOD and LOW. Additionally an upregulation of inflammatory responses was observed in HIGH and LOW, but not MOD, although the exact molecular transducers that are involved are unclear. Conversely, the three different exercise sessions evoked remarkably similar alterations in the expression of circadian genes, phosphorylation of some key metabolic proteins and circulating cytokine concentrations despite the vastly different exercise stimuli. Leveraging bioinformatic tools, we provide valuable clinical insights about exercise-induced responses in aSAT, including the identification of gene groups that may be tightly linked with circulating cytokine concentrations during exercise and the possibility of changes in the proportions of different cell types in response to exercise. This work expands the understanding of molecular responses in aSAT triggered by a session of exercise that may contribute to the maintenance or remodelling of aSAT in regular exercisers, which, in turn, may be an important factor underlying the preserved cardio-metabolic health often attributed to regular exercise.

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

## Additional information

### Data availability statement

The raw fastq files of RNA-seq are available in GEO (Gene Expression Omnibus) under the accession number GSE303983.

Gene count and metadata are hosted on GitHub (https://github.com/ahnchi/3X-Study). All other datasets generated and analysed in this study are available from the corresponding author upon reasonable request.

R codes that were used for differential analysis and WGCNA are hosted on GitHub (https://github.com/ahnchi/3X-Study).

## Competing interests

The authors declare that they have no competing interests.

## Author contributions

S.C.J.P., C.F.B. and J.F.H. designed the study. C.A., T.Z., G.Y., T.R., O.K.C., S.E., S.J.G., S.M., R.S. and J.F.H. contributed to data acquisition. All authors have participated in data analysis and interpretation. C.A. and J.F.H. drafted the work. All authors have participated in revising the work and approved the final version of the manuscript.

## Funding

HHS | NIH | National Institute of Diabetes and Digestive and Kidney Diseases (NIDDK): Jeffrey F. Horowitz, R01DK131724; HHS | NIH | NIDDK | Division of Diabetes, Endocrinology and Metabolic Diseases (DEM): Jeffrey F. Horowitz, P30DK089503.

## Acknowledgements

We are grateful to acknowledge the contributions of the study participants. We also acknowledge the excellent technical assistance from the Clinical Genomics Centre at the Oklahoma Medical Research Foundation and all the members of the Substrate Metabolism Laboratory. This study was supported by The National Institutes of Health (R01DK131724, P30DK089503, JFH). The funders had no role in the study design, data collection and analysis, decision to publish or preparation of the manuscript.

## Keywords

acute exercise, adipose tissue, exercise intensity

## Supporting information

Additional supporting information can be found online in the Supporting Information section at the end of the HTML view of the article. Supporting information files available:

**Peer Review History**

