## [Peer Review History · The Journal of Physiology]

Molecular responses in abdominal subcutaneous adipose tissue after a session of endurance exercise: effects of exercise intensity

Cheehoon Ahn, Tao Zhang, Thomas Rode, Gayoung Yang, Olivia K Chugh, Sierra Ellis, Sophia J Ghayur, Shriya Mehta, Ryan Salzman, Hui Jiang, Stephen CJ Parker, Charles F. Burant, and Jeffrey F. Horowitz

DOI: 10.1113/JP289339

Corresponding author(s): Jeffrey Horowitz (jeffhoro@umich.edu)

The following individual(s) involved in review of this submission have agreed to reveal their identity: Thomas Åskov Pedersen (Referee #2); Sean Williams (Referee #3)

Review Timeline:

Submission Date:	16-Jun-2025
Editorial Decision:	17-Jul-2025
Revision Received:	19-Aug-2025
Editorial Decision:	08-Sep-2025
Revision Received:	15-Sep-2025
Accepted:	22-Sep-2025

Senior Editor: Karyn Hamilton

Reviewing Editor: Jørn Helge

Transaction Report:

Dear Dr Horowitz,

Re: JP-RP-2025-289339 "**Acute session of three endurance exercise intensities alters subcutaneous adipose tissue transcriptome in regular exercisers**" by Cheehoon Ahn, Tao Zhang, Thomas Rode, Gayoung Yang, Olivia K Chugh, Sierra Ellis, Sophia J Ghayur, Shriya Mehta, Ryan Salzman, Hui Jiang, Stephen CJ Parker, Charles F. Burant, and Jeffrey F. Horowitz

Thank you for submitting your manuscript to The Journal of Physiology. It has been assessed by a Reviewing Editor and by 3 expert referees and we are pleased to tell you that it is potentially acceptable for publication following satisfactory major revision.

REVISION CHECKLIST:

We look forward to receiving your revised submission.

Yours sincerely,

Karyn Hamilton
Senior Editor
The Journal of Physiology

REQUIRED ITEMS

- Include a Key Points list in the article itself, before the Abstract.

- Author photo and profile. First or joint first authors are asked to provide a short biography (no more than 100 words for one author or 150 words in total for joint first authors) and a portrait photograph. These should be uploaded and clearly labelled together in a Word document with the revised version of the manuscript. See Information for Authors for further details.

- You must start the Methods section with a paragraph headed Ethical Approval. If experiments were conducted on humans, confirmation that informed consent was obtained, preferably in writing, that the studies conformed to the standards set by the latest revision of the Declaration of Helsinki and that the procedures were approved by a properly constituted ethics committee, which should be named, must be included in the article file. If the research study was registered (clause 35 of the Declaration of Helsinki), the registration database should be indicated, otherwise the lack of registration should be noted as an exception (e.g. The study conformed to the standards set by the Declaration of Helsinki, except for registration in a database). For further information see: <https://physoc.onlinelibrary.wiley.com/hub/human-experiments>.

- The reference list must be in alphabetical order, rather than numbered, to comply with our Journal format.

- Please upload separate high-quality figure files via the submission form.

- Your paper contains Supporting Information of a type that we no longer publish, including supplementary tables and figures. Any information essential to an understanding of the paper must be included as part of the main manuscript and figures. The only Supporting Information that we publish are video and audio, 3D structures, program codes and large data files. Your revised paper will be returned to you if it does not adhere to our Supporting Information Guidelines.

- Papers must comply with the Statistics Policy: https://jp.msubmit.net/cgi-bin/main.plex?form_type=display_requirements#statistics.

In summary:

- If n {less than or equal to} 30, all data points must be plotted in the figure in a way that reveals their range and distribution.

A bar graph with data points overlaid, a box and whisker plot or a violin plot (preferably with data points included) are acceptable formats.

- If $n > 30$, then the entire raw dataset must be made available either as supporting information, or hosted on a not-for-profit repository, e.g. FigShare, with access details provided in the manuscript.

- 'n' clearly defined (e.g. x cells from y slices in z animals) in the Methods. Authors should be mindful of pseudoreplication.

- All relevant 'n' values must be clearly stated in the main text, figures and tables.

- The most appropriate summary statistic (e.g. mean or median and standard deviation) must be used. Standard Error of the Mean (SEM) alone is not permitted.

- Exact p values must be stated. Authors must not use 'greater than' or 'less than'. Exact p values must be stated to three significant figures even when 'no statistical significance' is claimed.

- Please include an Abstract Figure file, as well as the Figure Legend text within the main article file. The Abstract Figure is a piece of artwork designed to give readers an immediate understanding of the research and should summarise the main conclusions. If possible, the image should be easily 'readable' from left to right or top to bottom. It should show the physiological relevance of the manuscript so readers can assess the importance and content of its findings. Abstract Figures should not merely recapitulate other figures in the manuscript. Please try to keep the diagram as simple as possible and without superfluous information that may distract from the main conclusion(s). Abstract Figures must be provided by authors no later than the revised manuscript stage and should be uploaded as a separate file during online submission labelled as File Type 'Abstract Figure'. Please also ensure that you include the figure legend in the main article file. All Abstract Figures should be created using BioRender. Authors should use The Journal's premium BioRender account to export high-resolution images. Details on how to use and access the premium account are included as part of this email.

EDITOR COMMENTS

Reviewing Editor:

Ethics Concerns:

The approval number is missing in the manuscript.

Reviewer one suggest that one statistical approach was not properly chosen and in fact incorrect. I would like to see the authors' response.

This is an interesting and well-written paper addressing the different effects of acute exercise modalities. The two reviewers have some concerns about the methodology and the description and the interpretation of data, which require serious attention by the authors. Furthermore, both reviewers note lack of information on the ethic and one a statistical approach.

Please also see 'Required Items' above.

Senior Editor:

Comments for Authors to ensure the paper complies with the Statistics Policy (Required):

If you choose to provide a revised manuscript please visit The Journal's Statistics Policy to ensure full compliance. For example, we noted that precise p-values are not consistently included and, in some places, SEM is used rather than SD. Thank you.

Comments to the Author:

Thank you for submitting your manuscript for consideration by The Journal of Physiology. As part of the peer review process, we recruited two Referees with expertise in this field of study. Both Referees noted strengths of the study and provided very detailed feedback. This feedback included notes about some major concerns including some related to data analyses and interpretation. To provide extra feedback for you, we also secured a consultation by a statistical expert. If you believe you can satisfactorily address the Referees' concerns, we welcome a point-by-point response accompanying a revised manuscript. If you do revise the manuscript, please make certain to consult The Journal's statistics policy and also include the institutional ethics committee approval number for this study. Of course, revision does not guarantee acceptance and we certainly understand if you choose not to. Thank you and we appreciate your interest in The Journal of Physiology!

REFeree COMMENTS

Referee #1:

The present paper explores the transcriptional changes in abdominal subcutaneous adipose tissue (aSAT) acutely after different exercise bouts. Three different intensities (and durations) of cycling exercise, LOW, MOD and HIGH were performed and aSAT retrieved before and after exercise and subjected to bulk RNA seq. A very high number of differentially expressed genes are found and these are explored by different bioinformatic tools. Some additional measurements of blood parameters and a few staining are also included for comparison to the transcriptional changes.

The main message is that the acute exercise induces numerous (~15% of the transcriptome) and fast changes (90 min) in the aSAT. Furthermore, the changes are quite different between the three different exercise modes.

The experimental idea is sound and relevant. Also, the use of different bioinformatic tools to explore the results seems fit in most cases. The paper is well-written and the data in general nicely presented and discussed. However, as the results (at least to this reviewer) is quite surprising, see below, this put extra emphasis on the technical descriptions and here there is room for improvement.

Major comments

The primary outcome is the changes in the transcriptome shortly after an acute bout of exercise. Here, roughly the same number of differentially expressed genes are found in all three exercise types and also a similar number of up and down-regulated genes in all three cases. Furthermore, there is little overlap between the exercise types both in which genes and the direction. This raises a red flag as this is exactly what would be expected for a random selection. Therefore, I suggest adding much more details to the method section for differential expression (currently mainly mention the programs) and that this red flag is addressed in the discussion, including a more thorough physiological explanation (or attempt) for why there could be this divergent difference in how the aSAT respond to the three exercise types (section two in the discussion) and why this does not translate into the signaling pathways measured by WB.

"The primary aim of this study was to compare the acute effects of three exercise intensities on abdominal subcutaneous adipose tissue (aSAT) transcriptome in regular exercisers" (quote abstract). Then it is a major drawback that no direct comparisons are made between the transcriptomes of the three exercise types. Only indirect comparisons are made, e.g. comparing presence of significant GO-terms Fig 3B-C. An attempt for a direct comparison is made in Fig 3D-E by comparing only the DEGs between the different exercise types. However, this is not a statistically valid approach, as the DEGs are not randomly selected, but specifically selected as the highest/lowest values within the specific group. To make a valid comparison, all detected genes should be included, not only the DEGs.

The "Differential analysis, pathway analysis, WGCNA module membership of genes" are stated to be available at GitHub, but as these pages are not (yet) available, I have not been able to use this in the reviewing process. Likewise, The R code is stated to be available on GitHub and this might have complemented the very brief method section.

Also, the guidelines states "The Journal of Physiology requires that nucleic acid and protein sequences, microarray data and data obtained using high throughput sequencing techniques, which support the results in the manuscript, should be archived in an appropriate public database (see examples) and must be accessible without restriction from the date of publication. Exceptions may be granted at the discretion of the Editor e.g. for sensitive information that might compromise the anonymity of human subjects.

An entry name or accession number, together with a direct link, must be included within the Methods section.". No reference to a public link containing these data is provided in the manuscript.

A previous experiment from the same lab (ref 42) also performed transcriptome analysis on aSAT after an acute exercise bout, though in obese subjects and found only very few genes changed and "Finally, our findings indicate that low-volume,

high-intensity exercise evokes many similar adipose tissue transcriptomic responses to more conventional moderate-intensity exercise". This apparent discrepancy should be discussed.

Minor comments

Reference number for the ethics approval is missing.

How and from where were the subjects recruited, e.g. running club?

The subjects were engaged in endurance exercise, but which type of exercise, e.g. running or biking, and was it elite or recreational?

Line 77-84: It is not clear in the method whether the listed subject characteristics were the inclusion/exclusion criteria or summary observations after the enrollment.

Group assignment. The subjects are very well matched. But how was this achieved? Please elaborate on "using a counter-balanced approach" in line 89.

It is not fully clear what the subjects have been doing prior to the acute exercise bout. In Fig. 1B is written "2-day exercise wash-out". Add the given instructions to the subjects in the method section.

The location of the aSAT sampling is stated as "5-10 cm distal to the umbilicus" and "the other side of the umbilicus". Is that correctly stated? For comparable biopsies it would make more sense to take the biopsies left and right from the umbilicus, not below and above.

Line 140: Add amount of tissue used for RNA purification and amount of total RNA used for polyA enrichment.

Line 143-144: How many reads per sample? And was PCR duplicates taken into account (deduplication)?

Line 153-155: Which background was used for the enrichment analyses, all human genes (IMHO incorrect) or the genes used for statistics?

Line 179: "previously published single-nuclei (sn) RNAseq datasets of human adipose tissue (27-29)". Reference 28, Backdahl et al, is a spatial transcriptomics study, not snRNAseq.

Line 198: " centrifuged at 4{degree sign}C for 3 x 15 minutes at 15,000g". Was supernatant removed or transferred in between the centrifugations?

Line 199-200: "Protein concentration was assessed using the bicinchoninic acid method (#23225, Thermofisher) after removing the supernatant". Is this a mistake? You are measuring soluble proteins, so why use the pellet?

Line 236: Two-way ANOVA linear mixed model can be performed in so many different ways using R, so more details are needed, e.g. the model, R packages. Also, "Post-hoc Tukey test was applied when significant interaction effect was detected" but then where does the double crosses for "main effect of time" come from in figure S1?

Line 266-268: "1hPost values remained slightly, yet significantly higher than Pre in both LOW and MOD (both $p \leq 0.05$), but not in HIGH (Figure 2B)". In the figure significance is not indicated and why suddenly " $p \leq 0.05$ " when everything else is " $p < 0.05$ "?

Line 286: "Figure S2E" -> "Figure S2A".

Line 360-362: The methods and results for histological traits are missing.

Line 460: "Notably, many responses to acute exercise were different among our three exercise treatments. For example, cytokine production was downregulated by MOD and LOW but upregulated by HIGH. Conversely, oxidative phosphorylation and ribosome biogenesis were more enriched by MOD compared with LOW or HIGH, which is consistent with the findings from a previous exercise training study (38)". I'm not sure what you consider consistent with the findings of Ronn et al (38)? They used only one training regime (probably most similar to LOW), so they did not compare LOW, MOD, HIGH.

I appreciate the limitation section and generally agree. However, I think you underestimate the changes in clock genes, "However, given the relatively short time between biopsies (<3h), it would be surprising if the robust changes we observed were solely driven by these factors". For instance, NR1D1 likely peaks at the end of the experiment with an increase of maybe 2-fold every 90 min, that is 3-4-fold from Pre to Post (<https://www.nature.com/articles/s41598-019-39668-3>).

Table 1: Consider subdivision into sex, perhaps in a supplemental table. And add sports type if subjects are recruited from more than one.

Figure S2: Cytokine measurements are performed using a multiplex kit, so presumably all together. However, the sample size ranges from 10 to 15 between the different cytokines. The reason why up to one third of the values are missing should be provided

Referee #2:

The response to bouts of exercise and prolonged exercise training in adipose tissue is an interesting topic to study in order to understand the metabolic improvements and health benefits that happens in connection to exercise.

The manuscript is well written and the augmentation easy to follow.

The study represents a thorough analysis of the bulk transcriptomic response in adipose tissue to different exercise protocols with different intensities. The study includes a sufficient number of participants, The participant groups are well pheno-typed and well matched and the researchers use state of the art computational methods to decipher the main pathways and cell types that responds to the exercise in abdominal subcutaneous adipose tissue.

The findings highlighted after the general DEG, includes identification of regulators of the cellular circadian clock as responsive to exercise intervention. The study is not followed up with regards to examination of the down stream consequences for these interesting findings

The study is extensive, largely descriptive and provides the basis for several hypothesis regarding the effects of exercise on (abdominal) subcutaneous adipose tissue. In it self the data provides an important piece of the puzzle towards understanding the physiological beneficial effects of exercise.

The authors state that the analysis of responses to single bouts of exercise aids the understanding of training interventions. I agree with this and to highlight the contributions of this work to that understanding, the comparison or implications of the results in this paper to previous works on training effects in AT should be highlighted by more thoroughly in the discussion.

Specific comments:

If NR1D1 is the most regulated gene across conditions and the circadian regulators assessed are co-regulated, it is odd that this is not picked up computationally. Luckily, the authors still manually curated the expression data, even though the emphasis of this work is clearly on the computational methods and analyses. The fact that this finding was missed by the automated analysis is not discussed by the authors although they appear very competent to comment on this.

"Gene modules identified by WGCNA are linked to clinical traits"

Some data is difficult to find or is missing from the paper. E.G. It is difficult for the reader to find the quantification of the histological data on adipocyte size, Sirius red, CD14 and CD206 positive cells. It should be referred to in "Gene modules identified by WGCNA are linked to clinical traits" and should be included in the table or at least in supplementary data. If the authors have used data from other sources, they should be cited.

The idea behind this analysis is interesting (that analysis could be more robust if considering MEs rather than single genes) and could be substantiated by integrating more data e.g. the data available at the newly published "adiposetissue.org"

Figure 7 and Integration with single-cell data suggests cell-type dependent responses in aSAT by acute exercise:

I agree that different cells contribute differently to the response and that deconvolution methodology had made recent progress. The overall claims are fine. The claim on changes in cell population based on deconvolution is difficult for me to comprehend and could be followed up by more direct methods like histology. Although numbers of mobile cells like infiltrating macrophages can certainly be dynamic, I have difficulties to envision how numbers of adipocytes and vascular cells should change within hours.

In addition, two of the papers used are based on single nuclear RNA, whereas the last is based on spatial transcriptomics that, like bulk sequencing, assess the mRNA pool. It is not clear to me how this affects the analysis and this is not discussed. It would be nice if the authors could address this in a couple of sentences.

Discussion: Line 490: The mouse model described by Hunter et al. is adipocyte specific and does not remove NR1D1 from the entire adipose tissue.

The involvement of the erk pathway in adipogenesis is complicated and even though its tempting to speculate as the authors are doing, it is difficult to infer effects of a transient signalling event across a whole tissue to the effects on differentiation in a distinct population. Measures of signalling at higher cellular resolution is needed for this.

Referee #3 (statistics review):

At present, the 'Statistical Analysis' section is very brief and would benefit from a more detailed description of the methods used. The phrase "two-way ANOVA linear mixed model" is unclear and risks conflating traditional ANOVA with linear mixed-effects models. The authors should provide a precise description of the model structure, including the fixed and random effects. For example, they could write: "A two-way linear mixed-effects model was used with [factor 1] and [factor 2] as fixed effects, and [random effect, for example subject ID] as a random effect to account for repeated measures." It is also important that the authors describe how the key assumptions of the model were assessed, such as the normality and homogeneity of residuals.

There is a small redundancy in the current text, where the use of R is mentioned twice (lines 238 and 240). This should be streamlined, and the citation should follow a consistent format, such as: R Statistical Software (v4.1.2; R Core Team, 2021). The authors should also list and cite any key R packages used in the analysis, for example lme4.

END OF COMMENTS

We greatly appreciate the reviewers for their evaluation and comments on our manuscript. We have revised the manuscript to address the reviewers' comments, and we believe these revisions have greatly improved our manuscript. We have provided a point-by-point response to all the reviewers' specific comments below (our responses are in bold).

EDITOR COMMENTS

Reviewing Editor:

Ethics Concerns:

The approval number is missing in the manuscript.

We have added the ethics approval and clinicaltrials.gov reference numbers.

Reviewer one suggest that one statistical approach was not properly chosen and in fact incorrect. I would like to see the authors' response.

This is an interesting and well-written paper addressing the different effects of acute exercise modalities. The two reviewers have some concerns about the methodology and the description and the interpretation of data, which require serious attention by the authors. Furthermore, both reviewers note lack of information on the ethic and one a statistical approach.

We appreciate the editor's assessment that our manuscript is well-written and presents meaningful physiological insights. In response to the reviewers' concerns, we have revised the manuscript to clarify our RNA-seq analysis approach (Lines 167-187, 190-198) and address potential sources of confusion regarding the interpretation of our data (Lines 399-426, 542-547, 561-570, 573-595). We have also expanded the statistical methods section to provide a more detailed description of our models and covariates (Lines 322-339).

Please also see 'Required Items' above.

- **We have included the Key Points section in the manuscript, preceding the Abstract.**
- **Author photo and profile have been submitted as requested.**
- **Ethical approval, including the IRB registration number and ClinicalTrials.gov ID, has now been added at the beginning of the Methods section.**
- **Reference list is now in alphabetical order**

- **Our Figures are now uploaded as separate, high-quality figure files.**
- **All supplementary figures have been integrated into the main figures**
- **We have made every effort to comply with the journal's statistical reporting guidelines. Boxplots were used where appropriate, and individual data points have been overlaid on bar graphs. Mean (SD) is reported consistently throughout the manuscript. While we have reported exact p-values wherever possible (either in text or figure legends), due to the complexity of the study design (three groups, multiple time points) and the scale of data (e.g., high-throughput RNA-seq and downstream analyses), we used $p < 0.001$ for any results with smaller p-values. For RNA-seq downstream analyses-including gene set testing and WGCNA, all reported pathways meet statistical significance following multiple testing correction ($FDR < 0.05$), as described in the relevant Methods sections.**

Senior Editor:

Comments for Authors to ensure the paper complies with the Statistics Policy (Required):

As noted above, we have made every effort to align with The Journal's Statistics Policy. In our revised manuscript, standard deviation (SD) is used consistently throughout in place of SEM. We have also ensured that precise p-values are reported wherever feasible, with exceptions only in the case of highly significant values (e.g., $p < 0.001$), due to the complexity and scale of certain analyses.

REFEREE COMMENTS

Referee #1:

Major comments

1. The primary outcome is the changes in the transcriptome shortly after an acute bout of exercise. Here, roughly the same number of differentially expressed genes are found in all three exercise types and also a similar number of up and down-regulated genes in all three cases. Furthermore, there is little overlap between the exercise types both in which genes and the direction. This raises a red flag as this is exactly

what would be expected for a random selection. Therefore, I suggest adding much more details to the method section for differential expression (currently mainly mention the programs) and that this red flag is addressed in the discussion, including a more thorough physiological explanation (or attempt) for why there could be this divergent difference in how the aSAT respond to the three exercise types (section two in the discussion) and why this does not translate into the signaling pathways measured by WB.

We appreciate the reviewer's observation regarding the limited overlap of differentially expressed genes (DEGs) across exercise types and that the pattern of up- and down-regulated genes may be what would be expected for a random selection. For full transparency, as we began to analyze our RNAseq data, our initial differential expression analysis was conducted using DESeq2, which accounted for repeated measures (paired pre- and post-exercise samples) and group-specific effects using the model formula: $\sim \text{group} + \text{group}:\text{ID} + \text{group}:\text{condition}$, where group represents the exercise treatment arm, ID denotes participant identity, and condition captures time point (Pre or Post). Gene set testing results (that use all detected genes) using output from DESeq2 were included in the supplementary figures of our previous version. Given recommendations from our Biostatistics core, we explored a multivariate approach using mashr (Multivariate Adaptive Shrinkage in R), which is capable of jointly modeling effects across multiple conditions and was thought to potentially offer more statistical power in detecting shared patterns across groups. Mashr takes in the log2foldchange and adjusted p-value from DESeq2 and quantifies posterior effect (i.e., effect size) and local false sign rates (lfsr; statistical significance).

In the original draft of our manuscript, our mashr analysis was limited to DEGs pre-filtered by DESeq2 (adjusted $p < 0.05$). But in response to the reviewer's concern, we tried to reapply mashr to all genes without pre-filtering and compared the posterior effect sizes and lfsr to the DESeq2 results. Interestingly, we observed strong concordance between DESeq2 and mashr for the HIGH group, both in terms of log-fold change and statistical significance. However, results were less consistent for the LOW and MOD groups—mashr-identified DEGs showed low overlap with DESeq2-identified genes, and importantly, no gene sets met the FDR threshold ($FDR < 0.05$) in those groups. These discrepancies raised concerns about overfitting or perhaps insufficient signal in the LOW and MOD intensity groups. Therefore, given the discrepancies with our mashr analysis, along with the widespread use and interpretability of DESeq2 in RNA-seq studies, in our revised manuscript we chose to rely exclusively on DESeq2 results for all differential expression analyses. Accordingly, we have

updated our methods section to provide a more thorough description of the modeling approach and justification for using DESeq2.

In terms of interpretation, we acknowledge the reviewer's concern that the lack of overlapping DEGs across groups could be mistaken for random noise. However, we believe this divergence may reflect true biology rather than statistical artifact. Exercise type, intensity, and modality are known to elicit distinct physiological responses. It is plausible that adipose tissue, which integrates both local and systemic signals, may engage different transcriptional programs in response to different exercise stimuli. This interpretation is supported by our pathway-level findings (now included in the main figure), where gene set enrichment analyses suggest distinct biological processes are altered by each exercise type, even in the absence of consistent DEG overlap. Furthermore, we employed CAMERA (Wu & Smyth, 2012), a competitive gene set testing approach that accounts for inter-gene correlation, which is known to yield more robust and reliable pathway-level inferences than traditional GSEA. We are therefore confident that the observed transcriptional divergence reflects the varied biological response.

Regarding the apparent disconnect between transcriptomic and phosphosignaling outcomes, this is a recognized limitation of our study. While phosphoprotein measurements offer insight into upstream regulatory events, they were limited to a small number of selected phosphosites on selected proteins, and sampled at a single time point (1.5hr post-exercise). Transcriptional changes may follow different kinetics or reflect longer-term regulatory processes not captured by our phosphosignaling snapshot. Thus, while informative in parallel, we contend that transcriptome and phosphosignaling data should be interpreted separately, as noted in our discussion of study limitations.

2. "The primary aim of this study was to compare the acute effects of three exercise intensities on abdominal subcutaneous adipose tissue (aSAT) transcriptome in regular exercisers" (quote abstract). Then it is a major drawback that no direct comparisons are made between the transcriptomes of the three exercise types. Only indirect comparisons are made, e.g. comparing presence of significant GO-terms Fig 3B-C. An attempt for a direct comparison is made in Fig 3D-E by comparing only the DEGs between the different exercise types. However, this is not a statistically valid approach, as the DEGs are not randomly selected, but specifically selected as the highest/lowest values within the specific group. To make a valid comparison, all detected genes should be included, not only the DEGs.

We thank the reviewer for highlighting the importance of direct comparisons between exercise types. We acknowledge that our initial analytical framework—using mashr on top of DESeq2—may have caused confusion. Importantly, this

approach did include direct comparisons across exercise modalities. For example, differential analysis was performed using contrast statements (via DESeq2) such as `contrast = list("GROUPa.TimePointa", "GROUPb.TimePointb")`, allowing for direct pairwise comparisons (e.g., HIGH vs. MOD). Mashr was applied on top of these comparisons. However, because mashr was applied only to genes pre-filtered as differentially expressed by DESeq2, downstream gene set testing was necessarily limited to this subset. As a result, only indirect functional comparisons (e.g., overrepresentation analysis) were feasible in those earlier figures. In response to this concern and to improve transparency, we have removed mashr from the final analysis and now report all results using DESeq2 only. Figures 3I–N show direct comparisons between exercise groups using DESeq2 contrast results, and gene set testing was performed using all detected genes included in the statistical model, rather than only DEGs. We believe this updated approach is more directly and statistically appropriate to address the primary aim of our study.

3. The "Differential analysis, pathway analysis, WGCNA module membership of genes" are stated to be available at GitHub, but as these pages are not (yet) available, I have not been able to use this in the reviewing process. Likewise, The R code is stated to be available on GitHub and this might have complemented the very brief method section.

Our revised methods section has been substantially expanded, and the R code for basic differential analysis and WGCNA is now publicly available from our GitHub repository (<https://github.com/ahnchi/3X-Study>). While the raw clinical data are not shared, the association results between WGCNA-derived modules and clinical traits are provided.

4. Also, the guidelines states "The Journal of Physiology requires that nucleic acid and protein sequences, microarray data and data obtained using high throughput sequencing techniques, which support the results in the manuscript, should be archived in an appropriate public database (see examples) and must be accessible without restriction from the date of publication. Exceptions may be granted at the discretion of the Editor e.g. for sensitive information that might compromise the anonymity of human subjects.

Thank you for pointing this out. The raw fastq files have been submitted to GEO and are expected to be publicly available in October 2025. The accession number is GSE303983, and the token for the reviewers is 'wnuzwysmldihnyl'

5. An entry name or accession number, together with a direct link, must be included within the Methods section." No reference to a public link containing these data is provided in the manuscript.

The accession number is now provided in the methods with an appropriate link.

6. A previous experiment from the same lab (ref 42) also performed transcriptome analysis on aSAT after an acute exercise bout, though in obese subjects and found only very few genes changed and "Finally, our findings indicate that low-volume, high-intensity exercise evokes many similar adipose tissue transcriptomic responses to more conventional moderate-intensity exercise". This apparent discrepancy should be discussed.

We thank the reviewer for this insightful comment. While both our study and Ludzki et al (Ludzki et al., 2021) investigated the acute effects of exercise on aSAT transcriptomes, several key methodological and biological differences likely account for the discrepancy in the number of DEGs reported. First, Ludzki et al. employed Lexogen QuantSeq 3' mRNA-seq (single-end, 100 bp), which captures only the 3' ends of transcripts. In contrast, our study used full-length, paired-end poly(A)-selected RNA-seq (PE150), providing greater coverage, sensitivity, and quantification accuracy. Second, their sequencing depth (~10 million reads, detecting 12,077 genes) was substantially lower than ours (~28.5 million reads on average, with over 17,000 genes retained after filtering). Third, we used different statistical frameworks: Ludzki et al. applied limma-voom, whereas we used DESeq2. Additionally, as the reviewer noted, the study populations differed: Ludzki et al. recruited participants with obesity, while ours consisted of adults without obesity who were regular exercisers.

Few studies have examined the effects of acute exercise on the human aSAT transcriptome, and to our knowledge, Fabre et al (Fabre et al., 2018) represents the most comprehensive study to date. In that work, sedentary adults without obesity underwent an acute aerobic exercise bout (15min, 75–85% VO₂max), and 3,882 genes were found to be differentially expressed in aSAT 4 hours post-exercise. Notably, when the same individuals repeated the exercise protocol after a 6-week training intervention, the number of differentially expressed genes decreased to 349. Although the study design and population differ from ours, these findings highlight the variability in transcriptomic responsiveness and underscore the complexity of interpreting adipose tissue gene expression following acute exercise.

While we agree this discrepancy is worth noting, the many methodological and biological differences between studies make a direct comparison challenging. For this reason, we have opted not to expand this discussion further in the main text. Instead, we now reference Fabre et al. in the Methods section to justify our recruitment of regular exercisers and acknowledge that training status can significantly influence transcriptomic responses to acute exercise (lines 94-97).

Minor comments

1. Reference number for the ethics approval is missing.

We have added the reference number for ethics approval and registration number for clinicaltrials.gov.

2. How and from where were the subjects recruited, e.g. running club?

Subjects were recruited from social media flyers and [UMHealthResearch.org](https://umhealthresearch.org).

3. The subjects were engaged in endurance exercise, but which type of exercise, e.g., running or biking, and was it elite or recreational?

All participants were regularly exercising at a recreational level. Of the 45 participants, 42 engaged in either running, cycling, or both (28 runners, 3 cyclists, and 11 who did both). The remaining three participants participated in tennis, kickboxing, or other forms of cardiovascular exercise. Thirteen participants also incorporated recreational-level resistance training into their routines. We have now added this information to *Participants* in the Methods.

4. Line 77-84: It is not clear in the method whether the listed subject characteristics were the inclusion/exclusion criteria or summary observations after the enrollment.

We agree the description in our original manuscript was not clear. The participant characteristics described here in the methods section were part of the eligibility criteria. We have now revised the text to state this clearly (*Participants* in the Methods).

5. Group assignment. The subjects are very well matched. But how was this achieved? Please elaborate on "using a counter-balanced approach" in line 89.

Thank you for your comment regarding the group assignment strategy. Group allocation was not randomized, but rather implemented through a manual counterbalancing approach. Specifically, as each subject was recruited, the study team reviewed their baseline characteristics (age, sex, BMI, body weight, VO₂peak) and assigned them to a group to help maintain similar distributions

across the intervention arms. This method allowed for reasonably balanced groups with respect to key variables of interest. We have now clarified this approach in the Methods (“Group assignment”).

6. It is not fully clear what the subjects have been doing prior to the acute exercise bout. In Fig. 1B is written "2-day exercise wash-out". Add the given instructions to the subjects in the method section.

We have now updated the Methods to state that all participants were instructed to refrain from exercise 48hours before the experimental trial. Their compliance was confirmed by the study coordinator.

7. The location of the aSAT sampling is stated as "5-10 cm distal to the umbilicus" and "the other side of the umbilicus". Is that correctly stated? For comparable biopsies it would make more sense to take the biopsies left and right from the umbilicus, not below and above.

The biopsies were taken from either left or the right side of the umbilicus for pre-exercise and for post, it was taken from the other side. While ‘distal’ does not mean below, we added ‘left or right’ to avoid confusion (*Experimental trial* in the Methods).

8. Line 140: Add amount of tissue used for RNA purification and amount of total RNA used for polyA enrichment.

We thank the reviewer for this suggestion. A total of 100 mg of frozen adipose tissue was used for RNA extraction. We submitted 20 µg of total RNA per sample to the Oklahoma Medical Research Foundation core. However, the exact amount of RNA used for poly(A) enrichment during library preparation was not tracked. We have now substantially expanded the RNA-seq Methods section to include these details.

9. Line 143-144: How many reads per sample? And was PCR duplicates taken into account (deduplication)?

On average, the reads were 28.5 million per sample. Deduplication is not a default step for RNA-seq data without UMI. FastQC was completed before rSeq, which reported that duplication was not a concern.

10. Line 153-155: Which background was used for the enrichment analyses, all human genes (IMHO incorrect) or the genes used for statistics?

For enrichment analysis, the background was set to genes (n=17466) that were used for statistics.

11. Line 179: "previously published single-nuclei (sn) RNAseq datasets of human adipose tissue (27-29)". Reference 28, Backdahl et al, is a spatial transcriptomics study, not snRNAseq.

We apologize for this oversight. We have corrected the description in the Methods ("*Cell-type enrichment and deconvolution*").

12. Line 198: " centrifuged at 4C for 3 x 15 minutes at 15,000g". Was supernatant removed or transferred in between the centrifugations?

The supernatants were transferred to new tubes after each spin. This was done to reduce lipid contamination. We have now added this detail in the Methods ("*Targeted immunoblots*").

13. Line 199-200: "Protein concentration was assessed using the bicinchoninic acid method (#23225, Thermofisher) after removing the supernatant". Is this a mistake? You are measuring soluble proteins, so why use the pellet?

Thank you for pointing this out. This was an oversight. We have corrected the text to state that the cleared supernatant was used for BCA.

14. Line 236: Two-way ANOVA linear mixed model can be performed in so many different ways using R, so more details are needed, e.g. the model, R packages. Also, "Post-hoc Tukey test was applied when significant interaction effect was detected" but then where does the double crosses for "main effect of time" come from in figure S1?

We have expanded the Methods section to provide greater detail on the statistical approaches used, including the exact models and R packages applied. Regarding Figure 2A (previously S1A), we observed a significant main effect of time at the 'Mid' time point. Follow-up pairwise comparisons revealed modality-specific time effects (Pre vs. MID: LOW, $p=0.032$; MOD, $p=0.035$; HIGH, $p=0.023$). While the original version reported only the main effect of time for simplicity—given that all groups showed significance—we have now included these pairwise results in the revised figure legend for completeness and clarity. Additionally, in our initial draft of the manuscript, we did not provide details regarding the changes in plasma calcium concentration (which is used to correct for exercise-induced changes in plasma volume (Alis et al., 2015)). However, because the impact of changes in plasma volume are important for interpreting some of our outcomes, we have now added more details describing changes in plasma calcium concentration (lines 356-362).

15. Line 266-268: "1hPost values remained slightly, yet significantly higher than Pre in both LOW and MOD (both $p \leq 0.05$), but not in HIGH (Figure 2B)". In the figure significance is not indicated and why suddenly " $p \leq 0.05$ " when everything else is " $p < 0.05$ "?

We previously used $p \leq 0.05$ as their statistics were at the borderline. In the revised draft, we have removed this sentence.

16. Line 286: "Figure S2E" -> "Figure S2A".

Thank you for pointing out this error. This figure is now 3A.

17. Line 360-362: The methods and results for histological traits are missing.

We appreciate the reviewer for catching this oversight. We have now added a detailed description of the histology methods in the revised Methods section ("*Histology*") and included the corresponding summarized outcomes in Table 1.

18. Line 460: "Notably, many responses to acute exercise were different among our three exercise treatments. For example, cytokine production was downregulated by MOD and LOW but upregulated by HIGH. Conversely, oxidative phosphorylation and ribosome biogenesis were more enriched by MOD compared with LOW or HIGH, which is consistent with the findings from a previous exercise training study (38)". I'm not sure what you consider consistent with the findings of Ronn et al (38)? They used only one training regime (probably most similar to LOW), so they did not compare LOW, MOD, HIGH.

We agree with the reviewer's point. Our intention was to highlight that the upregulation of oxidative phosphorylation observed with moderate-intensity exercise may align with previous findings by Rönner et al.; however, we recognize that the interpretation may have been overstated. We have deleted the original sentence noted by the reviewer, and revised the discussion to instead link the observed increase in oxidative phosphorylation with differences in exercise-induced energy expenditure (and mitochondrial metabolism) across groups.

19. I appreciate the limitation section and generally agree. However, I think you underestimate the changes in clock genes, "However, given the relatively short time between biopsies (<3h), it would be surprising if the robust changes we observed were solely driven by these factors". For instance, NR1D1 likely peaks at the end of the experiment with an increase of maybe 2-fold every 90 min, that is 3-4-fold from Pre to Post (<https://www.nature.com/articles/s41598-019-39668-3>).

We agree with the reviewer that many core clock genes—including NR1D1—are known to follow natural circadian rhythms. Since our study design does not allow us to disentangle circadian influences from true exercise-induced changes, we recognize the importance of being cautious in both presentation and interpretation. To reflect this, we have revised the text to avoid overstating these findings (Lines 635, 712-715).

20. Table 1: Consider subdivision into sex, perhaps in a supplemental table. And add sports type if subjects are recruited from more than one.

We have updated Table 1 to present the data in a sex-stratified format.

21. Figure S2: Cytokine measurements are performed using a multiplex kit, so presumably all together. However, the sample size ranges from 10 to 15 between the different cytokines. The reason why up to one third of the values are missing should be provided

The reviewer is correct about the analytes being measured altogether. For some analytes, measured concentrations were below the detection threshold in a subset of participants (IL6: 5 in LOW, 1 in MOD, 2 in HIGH; IL10: 1 in LOW, 1 in HIGH) at one or more time points. These participants were excluded from statistical testing for the respective analyte. This clarification has now been added to the Methods section (“*Blood measurements*”) and figure legend.

Referee #2:

The response to bouts of exercise and prolonged exercise training in adipose tissue is an interesting topic to study in order to understand the metabolic improvements and health benefits that happens in connection to exercise.

The manuscript is well written and the augmentation easy to follow.

The study represents a thorough analysis of the bulk transcriptomic response in adipose tissue to different exercise protocols with different intensities. The study includes a sufficient number of participants, The participant groups are well pheno-typed and well matched and the researchers use state of the art computational methods to decipher the main pathways and cell types that responds to the exercise in abdominal subcutaneous adipose tissue.

The findings highlighted after the general DEG, includes identification of regulators of the cellular circadian clock as responsive to exercise intervention. The study is not followed up with regards to examination of the down stream consequences for these interesting findings

The study is extensive, largely descriptive and provides the basis for several hypothesis regarding the effects of exercise on (abdominal) subcutaneous adipose tissue. In it self the data provides an important piece of the puzzle towards understanding the physiological beneficial effects of exercise.

The authors state that the analysis of responses to single bouts of exercise aids the understanding of training interventions. I agree with this and to highlight the contributions of this work to that understanding, the comparison or implications of the results in this paper to previous works on training effects in AT should be highlighted by more thoroughly in the discussion.

We sincerely appreciate the reviewer's thorough and thoughtful assessment of our study. We have carefully considered the suggestions and have revised the manuscript to better highlight how our findings may contribute to understanding the effects of both acute and prolonged exercise on adipose tissue. For example, we expanded our discussion regarding the lack of acute inflammatory upregulation—particularly in the MOD group—observed in our cohort. This finding is consistent with previous work by Fabre et al., (Fabre et al., 2018) who reported blunted inflammatory responses to acute moderate-intensity exercise following a period of exercise training. Together, these findings suggest that exercise intensity and the person's training history may help dictate the extent of the immune and inflammatory responses to a session of exercise (lines 590-595).

Specific comments:

If NR1D1 is the most regulated gene across conditions and the circadian regulators assessed are co-regulated, it is odd that this is not picked up computationally. Luckily, the authors still manually curated the expression data, even though the emphasis of this work is clearly on the computational methods and analyses. The fact that this finding was missed by the automated analysis is not discussed by the authors although they appear very competent to comment on this.

We apologize if the data presentation was unclear. In the volcano plots comparing Post vs. Pre within each exercise group (Figure 3A–C), NR1D1 is among the top 20 differentially expressed genes and is labeled in all three groups, positioned in the top left corner. Additionally, we observed that other core clock genes—such as DBP, NR1D2, NAMPT, and CIART—were also among the top hits. These

unbiased findings led us to further investigate and highlight circadian gene expression in our analysis. We have clarified this rationale in the revised manuscript to better convey how our interpretation was guided by the data (lines 430-432).

"Gene modules identified by WGCNA are linked to clinical traits"

Some data is difficult to find or is missing from the paper. E.G. It is difficult for the reader to find the quantification of the histological data on adipocyte size, Sirius red, CD14 and CD206 positive cells. It should be referred to in "Gene modules identified by WGCNA are linked to clinical traits" and should be included in the table or at least in supplementary data. If the authors have used data from other sources, they should be cited.

We acknowledge that the description of our histological measurements was limited. We have now added the data for CD14, CD206, and Sirius red abundances in Table 1, which is now presented in sex-, group-stratified manner.

The idea behind this analysis is interesting (that analysis could be more robust if considering MEs rather than single genes) and could be substantiated by integrating more data e.g. the data available at the newly published "adiposetissue.org"

We appreciate the reviewer's suggestion and are aware of the adiposetissue.org resource. Since all clinical and adipose tissue measurements in our study were performed in the same individuals, we believe our correlation analyses offer more directly interpretable insights. Additionally, because adiposetissue.org does not currently include data related to exercise interventions, it is less applicable for addressing our primary research focus. That said, we recognize its value and hope that our findings may contribute to this resource in the future.

Figure 7 and Integration with single-cell data suggests cell-type dependent responses in aSAT by acute exercise:

I agree that different cells contribute differently to the response and that deconvolution methodology had made recent progress. The overall claims are fine. The claim on changes in cell population based on deconvolution is difficult for me to comprehend and could be followed up by more direct methods like histology. Although numbers of mobile cells like infiltrating macrophages can certainly be dynamic, I have difficulties to envision how numbers of adipocytes and vascular cells should change within hours.

We understand the reviewer's concern. Without direct assessments such as histology or flow cytometry, we cannot conclude that acute exercise induces rapid changes in absolute cell type abundances. However, it is important to note that our deconvolution analysis estimates relative cell-type proportions rather

than absolute counts. Thus, a decrease in the proportion of adipocytes or vascular cells could reflect an increased infiltration of macrophages, even if the total number of adipocytes or vascular cells remains unchanged. We have added a sentence that points out this caveat (lines 528-530, 693-694).

In addition, two of the papers used are based on single nuclear RNA, whereas the last is based on spatial transcriptomics that, like bulk sequencing, assess the mRNA pool. It is not clear to me how this affects the analysis and this is not discussed. It would be nice if the authors could address this in a couple of sentences.

We appreciate the reviewer's observation. While all datasets assess mRNA abundance, we acknowledge the differences in transcript coverage. Single-nucleus RNA-seq (snRNA-seq) captures the nuclear transcriptome, which can differ from cytoplasmic or whole-cell mRNA profiles, whereas spatial transcriptomics and bulk RNA-seq typically measure both nuclear and cytoplasmic mRNA. Despite these methodological differences, recent work has shown that snRNA-seq data can effectively capture key cell-type-specific transcriptional programs. We now include a brief statement in the Methods to acknowledge these technical differences and their potential impact on cell-type deconvolution (lines 227-232).

Discussion: Line 490: The mouse model described by Hunter et al. is adipocyte specific and does not remove NR1D1 from the entire adipose tissue.

We agree with the reviewer's point and have revised the text to replace "adipose tissue" with "adipocyte" to more accurately reflect the cited reference.

The involvement of the erk pathway in adipogenesis is complicated and even though its tempting to speculate as the authors are doing, it is difficult to infer effects of a transient signalling event across a whole tissue to the effects on differentiation in a distinct population. Measures of signalling at higher cellular resolution is needed for this.

We strongly agree with the reviewer's point. Our immunoblotting approach was limited to a small number of phosphoproteins and select phosphosites, and thus cannot comprehensively capture the complexity of phosphosignaling events in adipose tissue. While ERK signaling has been implicated in adipocyte differentiation and metabolism, we acknowledge that our data do not allow us to draw conclusions about cell-type-specific differentiation at the tissue level. We have revised the Discussion section accordingly to tone down our interpretation and highlight this limitation (lines 673-675).

Referee #3 (statistics review):

At present, the 'Statistical Analysis' section is very brief and would benefit from a more detailed description of the methods used. The phrase "two-way ANOVA linear mixed model" is unclear and risks conflating traditional ANOVA with linear mixed-effects models. The authors should provide a precise description of the model structure, including the fixed and random effects. For example, they could write: "A two-way linear mixed-effects model was used with [factor 1] and [factor 2] as fixed effects, and [random effect, for example subject ID] as a random effect to account for repeated measures." It is also important that the authors describe how the key assumptions of the model were assessed, such as the normality and homogeneity of residuals.

We appreciate the reviewer's comment and agree that the original description lacked clarity. We have now substantially expanded the Methods section to provide detailed information on the statistical and bioinformatics approaches used, including explicit descriptions of model structures, fixed and random effects, and assessment of model assumptions. We believe these revisions address the reviewer's concerns.

Fabre, O., Ingerslev, L. R., Garde, C., Donkin, I., Simar, D., & Barres, R. (2018). Exercise training alters the genomic response to acute exercise in human adipose tissue. *Epigenomics*, *10*(08), 1033-1050.

Ludzki, A. C., Schleh, M. W., Krueger, E. M., Taylor, N. M., Ryan, B. J., Baldwin, T. C., Gillen, J. B., Ahn, C., Varshney, P., & Horowitz, J. F. (2021). Inflammation and metabolism gene sets in subcutaneous abdominal adipose tissue are altered 1 hour after exercise in adults with obesity. *Journal of Applied Physiology*, *131*(4), 1380-1389.

Wu, D., & Smyth, G. K. (2012). Camera: a competitive gene set test accounting for inter-gene correlation. *Nucleic Acids Res*, *40*(17), e133.
<https://doi.org/10.1093/nar/gks461>

Dear Dr Horowitz,

Re: JP-RP-2025-289339R1 "**Acute session of three endurance exercise intensities alters subcutaneous adipose tissue transcriptome in regular exercisers**" by Cheehoon Ahn, Tao Zhang, Thomas Rode, Gayoung Yang, Olivia K Chugh, Sierra Ellis, Sophia J Ghayur, Shriya Mehta, Ryan Salzman, Hui Jiang, Stephen CJ Parker, Charles F. Burant, and Jeffrey F. Horowitz

Thank you for submitting your manuscript to The Journal of Physiology. It has been assessed by a Reviewing Editor and by 3 expert referees and we are pleased to tell you that it is potentially acceptable for publication following satisfactory major revision.

REVISION CHECKLIST:

We look forward to receiving your revised submission.

Yours sincerely,

Karyn Hamilton
Senior Editor
The Journal of Physiology

REQUIRED ITEMS

- Please include an Abstract Figure file, ***as well as the Figure Legend text within the main article file*** (we seem to be missing the legend). The Abstract Figure is a piece of artwork designed to give readers an immediate understanding of the research and should summarise the main conclusions. If possible, the image should be easily 'readable' from left to right or top to bottom. It should show the physiological relevance of the manuscript so readers can assess the importance and content of its findings. Abstract Figures should not merely recapitulate other figures in the manuscript. Please try to keep the diagram as simple as possible and without superfluous information that may distract from the main conclusion(s). Abstract Figures must be provided by authors no later than the revised manuscript stage and should be uploaded as a separate file during online submission labelled as File Type 'Abstract Figure'. Please also ensure that you include the figure legend in the main article file. All Abstract Figures should be created using BioRender. Authors should use The Journal's premium BioRender account to export high-resolution images. Details on how to use and access the premium account are included as part of this email.

EDITOR COMMENTS

Reviewing Editor:

Authors have very adequately and diligently answered and explained the missing points. However, a few new issues raised by reviewer 1 needs some final attention. Please make sure that legend for abstract figure is available.

Senior Editor:

Thank you for submitting your revised manuscript for continued consideration. Your careful responses and revisions largely satisfied the referee concerns, however a few final points remain that require your attention. In addition to addressing these last Referee concerns, please do also provide a legend for your abstract figure. I apologize for overlooking that omission on your last submission. We are looking forward to seeing your revised manuscript and appreciate your interest in The Journal of Physiology.

REFeree COMMENTS

Referee #1:

I commend the authors for their fine work with accommodating the original comments. In general, I think the responses have been handled quite satisfactory.

However, the new analyses and method descriptions have induced a few new comments that needs to be addressed.

Major comments:

I'm happy to see that the new DEG analyses produce more plausible results that does deviate from randomness. The most obvious is figure 6C (old 5C) where the different groups are not perfectly balanced anymore. The DESeq2 approach now seems reasonable and with the new method description and the code I also find the DEG comparison between the three groups (fig 4I-K) statistical sound.

However, I think the headings in figure 4I-K are misleading as the comparisons are strictly between the changes from pre to post. That is, even though for instance USP33 and FKBP14 have a positive log2foldchange in figure 4J, and therefore belong to the "153 up in HIGH", they are not increased from pre to post in HIGH but reduced in LOW. Consider changing the heading and make a note in the legend/results to avoid incorrect interpretation by the readers.

The reason for the missing values for cytokine plasma concentrations has now been explained, line 288. Samples below detection have been excluded from the statistical testing. However, this is an invalid approach. In that way samples with lower values are specifically removed from the test creating a bias. A more appropriate solution is to assign these samples a value of zero or the threshold value.

Minor comments:

Line 200, "retaining only those with 10 >= counts in at least the smallest group", is not in line with the R code. The filtering is based on at least 15 samples with 10 or more counts, regardless of group.

The method for the statistical test is still missing for figure 4I-K. From the code it can be seen that contrasts from the Pre-Post test are compared. This is an important information, e. g. the changes are tested and should be included in the method even though it can be found in the code link.

Line 209, figure 3 has been changed to figure 4.

Helping note for the R code person: DESeq does take the order into account and will use the first factor level as reference. As "Post" comes before "Pre" alphabetically all log2foldchanges will be in the wrong direction. In the code this is handled by changing the "condition" of all Pre samples to Post and vice versa, but this is a dangerous solution. The correct solution is to use `relevel, colData$condition <- relevel(colData$condition, ref = "Pre")`. In line 339 this seems to have been performed correctly for the non-RNAseq statistics.

Referee #2:

I am overall satisfied by the response put forward by the authors. The publication of the dataset and the methods used could positively impact the field.

Referee #3:

Thank you - my comments have been satisfactorily addressed.

END OF COMMENTS

We thank the Senior Editor for the continued opportunity to revise and improve our manuscript. We have revised our manuscript again according to Referee #1's final comments – and we provide a point-by-point response to each of these comments below. We have also added a legend for the Graphical Abstract under the 'FIGURE LEGENDS' section, as requested. Additionally, we have now modified the title of our manuscript to better align with the revised manuscript. Changes in the manuscript are highlighted in yellow.

REFEREE COMMENTS

Referee #1:

Major comments:

I'm happy to see that the new DEG analyses produce more plausible results that does deviate from randomness. The most obvious is figure 6C (old 5C) where the different groups are not perfectly balanced anymore. The DESeq2 approach now seems reasonable and with the new method description and the code I also find the DEG comparison between the three groups (fig 4I-K) statistical sound.

We are very grateful for the reviewer's thoughtful and constructive feedback throughout the review process. Below are our responses to the remaining comments.

However, I think the headings in Figures 4I-K are misleading as the comparisons are strictly between the changes from pre to post. That is, even though for instance USP33 and FKBP14 have a positive log₂foldchange in figure 4J, and therefore belong to the "153 up in HIGH", they are not increased from pre to post in HIGH but reduced in LOW. Consider changing the heading and make a note in the legend/results to avoid incorrect interpretation by the readers.

We appreciate the reviewer's insight and fully agree that the original headings for Figure panels 4I-K may be misleading. While the text interpretation was carefully worded to emphasize that these DEGs represent relative 'enrichment' in one group compared with another group (e.g., MOD vs HIGH), we recognize the figure headings need to be more clear. We have now revised the headings for figure panels 4I-K and we also modified relevant portions in the text and legends by using 'larger changes (Δ) in expression'.

The reason for the missing values for cytokine plasma concentrations has now been explained, line 288. Samples below detection have been excluded from the statistical testing. However, this is an invalid approach. In that way samples with lower values are specifically removed from the test creating a bias. A more appropriate solution is to assign these samples a value of zero or the threshold value.

We thank the reviewer for identifying this issue. We have now imputed missing values with the lowest measured value for each cytokine and have re-plotted the figures and re-run the statistical analysis accordingly. This approach has also been noted in the Methods. Importantly, this adjustment did not affect the overall conclusions. The updated statistics are reflected in the revised manuscript text.

Minor comments:

Line 200, "retaining only those with 10 >= counts in at least the smallest group", is not in line with the R code. The filtering is based on at least 15 samples with 10 or more counts, regardless of group.

Thank you for catching this discrepancy. We have corrected the method description to match the filtering procedure used in the pipeline (Lines 194–196).

The method for the statistical test is still missing for figure 4I-K. From the code it can be seen that contrasts from the Pre-Post test are compared. This is an important information, e. g. the changes are tested and should be included in the method even though it can be found in the code link.

We have now revised the text in the Methods to state that Wald tests were used for both within-group and between-group comparisons, with examples of the code that is used to extract the contrast of interest.

Line 209, figure 3 has been changed to figure 4.

Thanks for pointing this out. It has now been corrected.

Helping note for the R code person: DESeq does take the order into account and will use the first factor level as reference. As "Post" comes before "Pre" alphabetically all log2foldchanges will be in the wrong direction. In the code this is handled by changing the "condition" of all Pre samples to Post and vice versa, but this is a dangerous solution. The correct solution is to use relevel, `colData$condition <- relevel(colData$condition, ref = "Pre")`. In line 339 this seems to have been performed correctly for the non-RNAseq statistics.

Thank you for the suggestion. We have updated the R code in GitHub to ensure correct orientation of log2 fold changes and avoid confusion in future use of the pipeline.

Dear Dr Horowitz,

Re: JP-RP-2025-289339R2 "Molecular responses in abdominal subcutaneous adipose tissue after a session of endurance exercise: effects of exercise intensity" by Cheehoon Ahn, Tao Zhang, Thomas Rode, Gayoung Yang, Olivia K Chugh, Sierra Ellis, Sophia J Ghayur, Shriya Mehta, Ryan Salzman, Hui Jiang, Stephen CJ Parker, Charles F. Burant, and Jeffrey F. Horowitz

We are pleased to tell you that your paper has been accepted for publication in The Journal of Physiology.

Yours sincerely,

Karyn Hamilton
Senior Editor
The Journal of Physiology

If you would like to receive our 'Research Roundup', a monthly newsletter highlighting the cutting-edge research published in The Physiological Society's family of journals (The Journal of Physiology, Experimental Physiology, Physiological Reports, The Journal of Nutritional Physiology and The Journal of Precision Medicine: Health and Disease), please click this link, fill in your name and email address and select 'Research Roundup':
<https://www.physoc.org/journals-and-media/membernews>

- You can help your research get the attention it deserves! Check out Wiley's free Promotion Guide for best-practice recommendations for promoting your work at: www.wileyauthors.com/eeo/guide. You can learn more about Wiley Editing Services which offers professional video, design, and writing services to create shareable video abstracts, infographics, conference posters, lay summaries, and research news stories for your research at: www.wileyauthors.com/eeo/promotion.

EDITOR COMMENTS

Reviewing Editor:

There are no further issues. Very nice work.

Senior Editor:

Thank you for submitting your revised manuscript. We are pleased to accept it for publication in The Journal of Physiology.
Thank you for your interest in The Journal and Congratulations!

REFEREE COMMENTS

Referee #1:

Well done.